# Dynamic Expectation Maximization Algorithm for Estimation of Linear Systems with Colored Noise

**DOI:** 10.3390/e23101306

**Published:** 2021-10-05

**Authors:** Ajith Anil Meera, Martijn Wisse

**Affiliations:** Department of Cognitive Robotics, Faculty of Mechanical, Maritime and Materials Engineering, Delft Institute of Technology, 2628 CN Delft, The Netherlands; m.wisse@tudelft.nl

**Keywords:** free energy principle, system identification, linear time-invariant (LTI) systems, colored noise, state space models, robotics

## Abstract

The free energy principle from neuroscience has recently gained traction as one of the most prominent brain theories that can emulate the brain’s perception and action in a bio-inspired manner. This renders the theory with the potential to hold the key for general artificial intelligence. Leveraging this potential, this paper aims to bridge the gap between neuroscience and robotics by reformulating an FEP-based inference scheme—Dynamic Expectation Maximization—into an algorithm that can perform simultaneous state, input, parameter, and noise hyperparameter estimation of any stable linear state space system subjected to colored noises. The resulting estimator was proved to be of the form of an augmented coupled linear estimator. Using this mathematical formulation, we proved that the estimation steps have theoretical guarantees of convergence. The algorithm was rigorously tested in simulation on a wide variety of linear systems with colored noises. The paper concludes by demonstrating the superior performance of DEM for parameter estimation under colored noise in simulation, when compared to the state-of-the-art estimators like Sub Space method, Prediction Error Minimization (PEM), and Expectation Maximization (EM) algorithm. These results contribute to the applicability of DEM as a robust learning algorithm for safe robotic applications.

## 1. Introduction

The free energy principle (FEP) has emerged from neuroscience as a unifying theory of the brain [1] and has begun to guide the search for a brain-inspired learning algorithm for robots. Many attempts have been made in this direction, including the state and input observer [2,3], the adaptive controller for robot manipulators [4,5,6], the body perception and action scheme for humanoid robots [7], the robot navigation of ground robots [8] etc. However, the design of a parameter estimation algorithm for linear systems with colored noise remains unexplored. Since the design of an accurate parameter estimator for dynamic systems sits at the core of control systems and robotics, the reformulation of FEP into a brain-inspired estimation algorithm has an influential impact on the industry and applied robotics.

A wide range of estimators have been proposed in the literature for linear time-invariant (LTI) systems [9], including the blind system identification [10,11,12]. However, most of them assume the noises to be temporally uncorrelated (white noise), which is often violated in practice. This results in biased estimation for the least-square (LS)-based methods [13], and an inaccurate convergence for the iterative methods [14]. Although many attempts have been made to solve this problem, mainly through bias compensation methods [15,16], none of them perform state, input, parameter, and noise estimation for systems with colored noises [17]. The only method that does it is the Dynamic Expectation Maximization (DEM) [18] algorithm, which uses FEP to invert a highly nonlinear and hierarchical brain model from sensory data. However, the disconnect between neuroscience and control system literature hinders the wide use of this method for practical robotics applications. Although FEP-based tools have already been applied to practical robotics applications [2,3,4,5,6,7,8,19,20], there is a gap in the literature on the applications of DEM owing to the mathematical formidability of the theory and its lack of formalism in the control systems domain. Therefore, it is important to reformulate DEM for the control systems audience. While DEM from computational neuroscience focuses on emulating the brain’s perception through the hierarchical abstraction of a number of interacting non-linear dynamic systems, our work focuses on reformulating it into a blind system-identification algorithm for an LTI system with colored noise, which is a well-known challenge in robotics. In this attempt, we keep all the brain-related approximations intact, thereby aiming for a biologically plausible parameter estimation algorithm.

According to an FEP proposed by Karl Friston, the brain’s inference mechanism is a gradient ascent over its free energy, where free energy is the information-theoretic measure that bounds the brain’s sensory surprisal [21]. FEP emerges as a unified brain theory [22] by providing a mathematical description of brain functions [23]; unifying action and perception [24]; connecting physiological constructs like memory, attention, value, reinforcement, and salience [23]; and remaining consistent with Freudian ideas [25]. Similarities of FEP with reinforcement learning [26], neural networks [21,27], PID controller [28], Kalman Filter [2] and active learning [24] open up possibilities for biologically plausible parameter estimation algorithms. Although FEP emerged as a brain theory, the recent works have pushed the boundaries towards systems that survive over time, such as social and cultural dynamics. Notable works include the variational approach to culture [29], collective intelligence [30], cumulative cultural evolution [31], etc.

Numerous methods have been proposed based on the FEP framework. Predictive coding [32] models perception through a hierarchy of dynamical systems [27] with the brain’s priors at the top, minimizing the prediction error at each level of the hierarchy. Bayesian message passing algorithms [33,34] use similar ideas for belief propagation. Active inference [24,35] uses FEP to model the brain’s action and perception under one framework. On the perception side, there are two main type of methods to deal with dynamic systems: variational filtering and generalized filtering. Variational filtering [18,36] uses mean-field approximation (conditional independence between densities), whereas generalized filtering [37] does not. DEM [18] is a type of variational filtering that uses a Laplace approximation [38] (a fixed-form assumption for the conditional density of variables), whereas [36] uses ensemble dynamics to model the free form of the conditional densities. We focus on DEM for this work.

DEM is an FEP-based variational inference algorithm that models the brain’s inference process as a maximization of its free energy for state, input, parameter, and noise estimation from data. Although the method shows high similarity to the variational inference [39], the key difference is in the use of generalized coordinates, which enables DEM to track the evolution of the trajectory of states instead of just the point estimates. This renders DEM with the capability of gracefully handling colored noises, a feature that conventional point estimators such as the Kalman Filter (KF) lacks. The modeling of noise color as analytic using generalized coordinates results in an improved state estimation under colored noise for LTI systems [2,3] and for nonlinear filtering [40], which directly improves the parameter estimation accuracy, making DEM a topic of interest [20]. This work directly impacts various sub-domains of robotics community: input estimation to the industrial automation community for fault detection systems, state estimation to the control systems community, parameter estimation to the system identification community, and hyperparameter estimation to the signal processing community.

With this paper, we aim to present DEM to the robotics audience as a robot learning algorithm for the blind system identification of LTI systems with colored noise. We elaborate on various components of DEM that are most relevant to the robotics community: (1) the derivation of the free energy objectives from Bayesian principles, (2) the modeling of colored noise using generalized coordinates, and (3) the simplification of update rules for LTI systems with colored noise. We reformulate DEM for LTI systems into a form that is widely used in the robotics domain and use this mathematical formulations to prove that the estimation steps of DEM have theoretical guarantees of convergence [41]. In our prior work, we have discussed the stability conditions of our DEM-based linear state and input observer design [2]. This convergence guarantees and stability criteria are essential for robot safety while in operation and is of high relevance to the robotics community. Through extensive simulations on a range of random systems, we show that DEM is a competitive estimator when compared to other benchmarks in the control systems domain. The core contributions of the paper are:Reformulating DEM into an estimation algorithm for LTI systems with colored noise (Section 12).Proving that the estimator has theoretical guarantees of convergence for the estimation steps (Section 14).Proving through rigorous simulation that DEM outperforms the state-of-the-art system identification methods for parameter estimation under colored noise (Section 16).

## 2. Problem Statement

Consider the linear plant dynamics given in Equation (Equation 1), where A, B and C are constant system matrices, $x\in R^{n}$ is the hidden state, $v\in R^{r}$ is the input and $y\in R^{m}$ is the output.
(1)$$\begin{array}{c} \dot{x}=Ax+Bv+w, \end{array}\;\;\begin{array}{c} y=Cx+z. \end{array}$$Here, $w\in R^{n}$ and $z\in R^{m}$ represent the process and measurement noise, respectively. The notations of the plant are denoted in boldface, whereas its estimates are denoted in nonboldface letters. The noises in this paper are generated through the convolution of white noise with a Gaussian kernel. The use of colored noise is motivated by the fact that in robotics, the unmodelled dynamics and the non-linearity errors can enter the plant dynamics through the noise terms, thereby violating the white noise assumption in practice [20].

The goal of this paper is to reformulate DEM for an LTI system such that given the output of the system y, the estimator computes the associated states *x*, inputs *v*, parameter θ containing the matrices A,B and *C*, hyperparameters λ that model the noise precision $\left(\Pi =e^{\lambda }\right)$, and the uncertainties of all its estimates $\left(\Sigma^{x},\Sigma^{v},\Sigma^{\theta },\Sigma^{\lambda }\right)$, with the help of the prior (learned) knowledge encoded in the robot brain, such that the estimate best predicts the data. The schematic of the proposed robot brain’s inference process is given in Figure 1.

## 3. Preliminaries

To reformulate DEM into an estimation algorithm for LTI systems, this section introduces the key concepts and terminologies that are familiar to the FEP audience.

### 3.1. Generative Model

The generative model (plant model) is the robot brain’s estimate of the generative process in the environment that generated data. The robot brain infers this model via model evidencing from the measurement data. The key idea behind DEM to deal with colored noise is to model the trajectory of the time-varying components (states, for example) using generalized coordinates. The use of generalized coordinates is new to the control systems literature and is different from the familiar definition in classical mechanics. In mechanics, it is the minimum number of independent coordinates that define the configuration of a system, whereas in DEM, it is the vector defining the motion of a point using its higher derivatives. In DEM, the emphasis is on tracking the trajectories of states, inputs and outputs instead of their point estimates. The states are expressed in generalized coordinates using its higher-order derivatives, i.e., $\tilde{x}={\left[x\;x^{\prime }\;x^{\prime \prime }\;\ldots \right]}^{T}$. The generalized state vector $\tilde{x}$ with an order of generalized motion of *p* will have p+1 terms, with the copy of the state vector as the first term, followed by its *p* derivatives. The variables in generalized coordinates are denoted by a tilde, and its components (higher derivatives) are denoted by primes. The evolution of states is written as:(2)$$\begin{array}{cc} \; & x^{\prime }=Ax+Bv+w\\ \; & x^{\prime \prime }=Ax^{\prime }+Bv^{\prime }+w^{\prime }\\ & \ldots \end{array}\;\;\begin{array}{cc} \; & y=Cx+z\\ \; & \dot{y}=Cx^{\prime }+z^{\prime }\\ & \ldots \end{array}$$The colored noises are modeled such that the covariance of noise derivatives $\tilde{z}={\left[z,z^{\prime },z^{\prime \prime },\ldots \right]}^{T}$ and $\tilde{w}={\left[w,w^{\prime },w^{\prime \prime },\ldots \right]}^{T}$ are well defined (to be explained in Section 3.3). The generative model representing the system is compactly written as:(3)$$\begin{array}{cc} \; & \dot{\tilde{x}}=D^{x}\tilde{x}=\tilde{A}\tilde{x}+\tilde{B}\tilde{v}+\tilde{w} \end{array}\;\;\begin{array}{cc} \; & \tilde{y}=\tilde{C}\tilde{x}+\tilde{z} \end{array}$$
where $D^{x}={\left[\begin{array}{ccccc} 0 & 1 & & & \\ & 0 & 1 & & \\ & & . & . & \\ & & & 0 & 1\\ & & & & 0 \end{array}\right]}_{\left(p+1)\times (p+1\right)}⊗I_{n\times n}$, performs the block derivative operation, equivalent to shifting up all components in generalized coordinates by one block. A similar definition holds for $D^{v}$ (appears later) with size r(d+1)×r(d+1), where *p* and *d* are the order of generalized motion of states and inputs, respectively. Here, $\tilde{A}=I_{p+1}⊗A$, $\tilde{B}=I_{p+1}⊗B$, and $\tilde{C}=I_{p+1}⊗C$, where ⊗ is the Kronecker tensor product.

### 3.2. Parameters and Hyperparameters

To simplify the parameter estimation steps of DEM for LTI systems (Section 13.2) and to facilitate the convergence proof (Section 14), we introduce an alternative generative model which is the direct reformulation of Equation (Equation 3) given by:(4)$$\begin{array}{c} \dot{\tilde{x}}=M\theta +\tilde{w},\;\tilde{y}=N\theta +\tilde{z},\;\theta =\left[\begin{array}{c} vec(A^{T})\\ vec(B^{T})\\ vec(C^{T}) \end{array}\right] \end{array}$$
where
(5)$$M=\left[\begin{array}{ccc} I_{n}⊗x^{T} & I_{n}⊗v^{T} & I_{n}⊗O_{1\times m}\\ I_{n}⊗x^{\prime T} & I_{n}⊗v^{\prime T} & I_{n}⊗O_{1\times m}\\ \ldots & \ldots & \ldots \end{array}\right],N=\left[\begin{array}{ccc} I_{n}⊗O_{1\times n} & I_{n}⊗O_{1\times r} & I_{m}⊗x^{T}\\ I_{n}⊗O_{1\times n} & I_{n}⊗O_{1\times r} & I_{m}⊗x^{\prime T}\\ \ldots & \ldots & \ldots \end{array}\right].$$Throughout the paper, θ represents the set of parameters, whereas $\lambda =\left[\begin{array}{c} \lambda^{z}\\ \lambda^{w} \end{array}\right]$ represents the hyperparameters that define the precision matrices (inverse covariance matrices) of the observation noise and the process noise. For noise modelling, we parametrize the noise precisions using an exponential relation with the hyperparameters, given by [18]:(6)$$\Pi^{w}\left(\lambda^{w}\right)=e^{\lambda^{w}}\Omega^{w},\;\Pi^{z}\left(\lambda^{z}\right)=e^{\lambda^{z}}\Omega^{z},$$
where $\Omega^{w}$ and $\Omega^{z}$ represent constant matrices encoding how different noises are correlated. Here $\Pi^{w}$ and $\Pi^{z}$ are the inverse covariances $\left\{{\left(\Sigma^{w}\right)}^{-1},\;{\left(\Sigma^{z}\right)}^{-1}\right\}$ or precisions of the noises. This parameterization ensures the selection of positive definite noise precision matrices through hyperparameter updates. We assume zero cross-correlation between *w* and *z*. We also assume that θ and λ are time-invariant.

### 3.3. Colored Noise

The next step towards handling the colored noise is to embed the noise correlation between different components in the generalized noises $\tilde{w}$ and $\tilde{z}$ given in Equation (Equation 3). DEM uses generalized coordinates, which models a correlation between noise derivatives through the temporal precision matrix *S* (inverse of covariance matrix). The generalized noise correlation is assumed to be due to a Gaussian filter, where *S* can be calculated as [18]:(7)$$S\left(\sigma^{2}\right)={\left[\begin{array}{cccc} 1 & 0 & -\frac{1}{2\sigma^{2}} & ..\\ 0 & \frac{1}{2\sigma^{2}} & 0 & ..\\ -\frac{1}{2\sigma^{2}} & 0 & \frac{3}{4\sigma^{4}} & ..\\ .. & .. & .. & .. \end{array}\right]}_{\left(p+1)\times (p+1\right)}^{-1}$$
where σ (from the Gaussian kernel) denotes the noise smoothness level. While $\sigma^{2}\to 0$ denotes white noise, nonzero $\sigma^{2}$ denotes colored noise. The covariance between noise derivatives increases exponentially with the order of noise derivatives. Simulations show that derivatives above 6 can be neglected [18]. The generalized noise precision matrices are given by ${\tilde{\Pi }}^{w}=S\left(\sigma^{2}\right)⊗\Pi^{w}$ and ${\tilde{\Pi }}^{z}=S\left(\sigma^{2}\right)⊗\Pi^{z}$, where $\Pi^{w}$ and $\Pi^{z}$ are the inverse noise covariances.

### 3.4. Generalized Motion of the Outputs and Noises

The generalized motion of output $\tilde{y}$ is practically expensive and inaccessible to robotics systems. Moreover, most sensors such as encoders operate with discrete measurements. Therefore, as a pre-processing step for estimation (refer to Figure 1), $\tilde{y}$ should be computed from discrete measurements. The goal is to first express the discrete measurements as a function of output derivatives and then invert the function to compute the derivatives from the discrete measurements. Given the *p* temporal derivatives $\tilde{y}$ at time *t*, the *p* output sequence surrounding y can be approximated using Taylor series as follows [18]:(8)$$\hat{y}=\left[\begin{array}{c} ..\\ y(t-dt)\\ y(t)\\ y(t+dt)\\ .. \end{array}\right]=\left(Y⊗I_{m}\right)\tilde{y},\;Y_{ij}=\frac{{\left[\left(i-ceil\left(\frac{p+1}{2}\right)\right)dt\right]}^{j-1}}{(j-1)!},$$
where i,j=1,2,…,p+1, ceil(.) is the smallest integer function and $\hat{y}$ has the size m(p+1)×1. Therefore, generalized motion of output $\tilde{y}$ at time *t* is:(9)$$\tilde{y}=\left(Y^{-1}⊗I_{m}\right)\hat{y}.$$Using $\tilde{y}$ embeds more temporal information about the plant into the data in the form of conditional trajectories, with the disadvantage of a time latency of $\frac{p}{2}dt$ in estimation. For robotic systems with high sampling rates, this latency in estimation is negligible [3,20]. The next section employs this generalized output along with the generative model for observer designs.

### 3.5. Notations and Conventions

Throughout the paper, the superscript notation will be used to represent the quantity being referred to, and the subscript notation will be used to represent the derivative operation. For example, ${\tilde{\Pi }}_{\lambda }^{w}$ represents the derivative of the generalized precision of the process noise *w* with respect to the hyperparameters λ. The tilde operator $\left(\tilde{x}\right)$ is used to represent a quantity in its generalized coordinates, whereas the bar operator $\left(\bar{F}\right)$ is used to represent the time integral of a quantity. All the probability distributions will be represented by the p(.) notation, whereas its expectation will be represented by the 〈p(.)〉 notation.

## 4. Free Energy Objectives

With the preliminaries in place, we can build up towards the complete DEM algorithm. Eventually, in Section 8 and Section 12, we will see that it is an optimization algorithm that finds the best estimates for states, inputs, parameters, and hyperparameters for given measurement data. This result is achieved by optimizing two objective functions, which are the core objectives under the entire Free Energy Principle: the free energy and the free action [1]. Here we derive and simplify this objective.

The derivation starts from the fundamentals of Variational Inference (VI) [39]. In VI, the posterior distribution p(ϑ/y) of parameter ϑ, given the measurement *y*, is expressed as:(10)$$p\left(\vartheta /y\right)=\frac{p(y/\vartheta )p(\vartheta )}{p(y)}=\frac{p(\vartheta ,y)}{p(y)}=\frac{p(\vartheta ,y)}{\int p(\vartheta ,y)d\vartheta }.$$However, the marginalization over ϑ to calculate p(y) is often intractable because the search space of ϑ is large. A widely used technique is to introduce a variational distribution q(ϑ) known as the recognition density, which acts as an approximate representation of the posterior distribution with q(ϑ)≈p(ϑ/y). A common method used among variational Bayes algorithms is to minimize the Kullback–Leibler (KL) divergence between both the distributions, defined as:(11)$$KL\left(q\left(\vartheta \right)||p\left(\vartheta /y\right)\right)={\langle \ln q\left(\vartheta \right)\rangle }_{q(\vartheta )}-{\langle \ln p(\vartheta /y)\rangle }_{q(\vartheta )},$$
where ${\langle .\rangle }_{q(\vartheta )}$ represents the expectation over q(ϑ). Substituting Equation (Equation 10) in Equation (Equation 11) and using ∫q(ϑ)dϑ=1 yields:(12)$$KL\left(q\left(\vartheta \right)||p\left(\vartheta /y\right)\right)={\langle \ln q\left(\vartheta \right)\rangle }_{q(\vartheta )}-{\langle \ln p(\vartheta ,y)\rangle }_{q(\vartheta )}+\ln p(y).$$The rearrangement of terms yield:(13)$$\begin{array}{cc} \ln p(y) & ={\langle \ln p(\vartheta ,y)\rangle }_{q(\vartheta )}-{\langle \ln q\left(\vartheta \right)\rangle }_{q(\vartheta )}+KL\left(q\left(\vartheta \right)||p\left(\vartheta /y\right)\right),\\ \; & =\underset{\mathrm{free}\;\mathrm{energy}}{\underset{︸}{{\langle U\left(\vartheta ,y\right)\rangle }_{q(\vartheta )}+H{\left(\vartheta \right)}_{q(\vartheta )}}}+KL\left(q\left(\vartheta \right)||p\left(\vartheta /y\right)\right), \end{array}$$
where lnp(y) is the log-evidence, U(ϑ,y)=lnp(ϑ,y) is the internal energy and $H{\left(\vartheta \right)}_{q(\vartheta )}=-{\langle \ln q\left(\vartheta \right)\rangle }_{q(\vartheta )}$ is the entropy of the density. The free energy term in Equation (Equation 13) is defined as the sum of an energy term and its entropy. It acts as the lower bound on the log-evidence because the KL divergence term KL(q(ϑ)||p(ϑ/y)) is always positive. The maximization of free energy minimizes the divergence term in Equation (Equation 13) because the log-evidence is independent of q(ϑ), thereby rendering the variational density q(ϑ) as a close approximation of p(ϑ/y). Therefore, the difficult evaluation of an intractable integral term in Equation (Equation 10) is converted into a much simpler optimization problem of maximizing the free energy. This reduces the problem of inference to a direct optimization of its free energy objectives and is the fundamental idea behind variation inference. The free energy term in Equation (Equation 13) is equivalent to the Evidence Lower Bound (ELBO) [39]. It can be simplified as:(14)$$\begin{array}{cc} F & ={\langle U\left(\vartheta ,y\right)\rangle }_{q(\vartheta )}+H{\left(\vartheta \right)}_{q(\vartheta )}=\int q\left(\vartheta \right)U\left(\vartheta ,y\right)d\vartheta -\int q\left(\vartheta \right)\ln q\left(\vartheta \right)d\vartheta \\ \; & =\int q(\vartheta )\ln p(\vartheta ,y)d\vartheta -\int q(\vartheta )\ln q(\vartheta )d\vartheta . \end{array}$$However, when the parameter set to be estimated includes both the time-variant and the time-invariant parameters, the free action is used as the objective function to be maximized. The free action is defined as the time integral of the free energy and is given by:(15)$$\bar{F}=\bar{V}\left(\vartheta \right)+\bar{H}\left(\vartheta \right)=\int {\langle U\left(\vartheta ,y\right)\rangle }_{q(\vartheta )}dt+\int H{\left(\vartheta \right)}_{q(\vartheta )}dt,$$
where $V\left(\vartheta \right)={\langle U\left(\vartheta ,y\right)\rangle }_{q(\vartheta )}$ is called the variational free energy (VFE).

The next sections will deal with two main assumptions used in DEM to simplify the free energy objectives, namely the Laplace approximation and the mean-field assumption. The aim is to derive the simplified free energy objectives for an LTI system under these assumptions.

## 5. Laplace Approximation

The first common approach to simplify the free energy objective is to assume the variational density q(ϑ) to be Gaussian in nature with variational parameters ϑ and $\Sigma^{\theta }$ as its mode and covariance, respectively, [38]. Here, the inverse of $\Sigma^{\vartheta }$ (denoted by $\Pi^{\vartheta }$ and known as the conditional precision), represents the confidence in estimation. The recognition density takes the following form:(16)$$q\left(\vartheta \right)=N\left(\vartheta :\mu^{\vartheta },\Sigma^{\vartheta }\right)=\frac{1}{\sqrt{{\left(2\pi \right)}^{n}\left|\Sigma^{\vartheta }\right|}}e^{-\frac{1}{2}{\left(\vartheta -\mu^{\vartheta }\right)}^{T}\Pi^{\vartheta }\left(\vartheta -\mu^{\vartheta }\right)}.$$There are two main advantages with this approximation:It simplifies the internal energy expression U(ϑ,y),It facilitates an easy computation of the conditional precision $\Pi^{\vartheta }$ (derived in Section 7.2) as the negative curvature of the internal energy at it’s mode $\mu^{\vartheta }$.Therefore, the main aim of this section is to simplify the expression for internal energy U(ϑ,y) using the Laplace approximation, for an LTI system with its states, inputs, and outputs expressed in generalized coordinates.

The internal energy in Equation (Equation 13) can be expressed as the sum of log-likelihood and prior terms as:(17)$$U\left(\vartheta ,y\right)=\ln p(\vartheta ,y)=\ln\underset{\begin{array}{c} \mathrm{generative}\\ \mathrm{model} \end{array}}{\underset{︸}{p(y/\vartheta )}}+\ln\underset{\mathrm{prior}}{\underset{︸}{p(\vartheta )}}.$$The parameter set ϑ includes two types of parameters:the states and inputs, which are time-varying and therefore expressed in generalized coordinates,the parameters and hyperparameters, which are time-invariant and not expressed in generalized coordinates.Equation (Equation 17) can be simplified by assuming the conditional independence of $\tilde{x}$ and $\tilde{v}$ with θ and λ. This factorization separates the deterministic quantities from the stochastic ones, thereby providing a separation of temporal scales. This is one of the core ideas behind DEM, which will be detailed in Section 6. With the redefinition of $\vartheta =\{\tilde{x},\tilde{v},\theta ,\lambda \}$, Equation (Equation 17) simplifies to:(18)$$\begin{array}{cc} U(\vartheta ,y)= & \ln p\left(y/\tilde{x},\tilde{v},\theta ,\lambda \right)+\ln p\left(\tilde{x},\tilde{v},\theta ,\lambda \right)\\ = & \ln p\left(y/\tilde{x},\tilde{v},\theta ,\lambda \right)+\ln p\left(\tilde{x}/\tilde{v},\theta ,\lambda \right)+\ln p\left(\tilde{v}\right)+\ln p\left(\theta \right)+\ln p\left(\lambda \right). \end{array}$$DEM combines the new sensory information *y* coming from the environment with the robot brain’s priors (refer to Figure 1) in a Bayesian fashion, through the internal energy U(ϑ,y) expression given in Equation (Equation 18). The next sections will deal with simplifying U(ϑ,y) and its action $\bar{U}$ by first modeling the probability distributions for the generative model and the priors.

### 5.1. Generative Model

The probability distribution $p(y/\tilde{x},\tilde{v},\theta ,\lambda )$, given in Equation (Equation 18), represents the generative model that predicts the output from the current parameter estimates. This probability can be assumed to be Gaussian-distributed, centered around the model’s output prediction $\tilde{C}\tilde{x}$ (from Equation (Equation 3)), with the same uncertainty as that of the measurement noise ${\tilde{\Sigma }}^{z}$. The distribution p(y/ϑ) becomes:(19)$$p\left(\tilde{y}/\vartheta \right)=\frac{1}{\sqrt{{\left(2\pi \right)}^{m(p+1)}\left|{\tilde{\Sigma }}^{z}\right|}}e^{-\frac{1}{2}{\left(\tilde{y}-\tilde{C}\tilde{x}\right)}^{T}{\tilde{\Pi }}^{z}\left(\tilde{y}-\tilde{C}\tilde{x}\right)}.$$Since the robot cannot directly measure $\tilde{x}$ in Equation (Equation 3), we track the motion of the generalized states through the approximation $\dot{\tilde{x}}={\tilde{x}}^{\prime }=D^{x}\tilde{x}$. The prediction for motion is $\tilde{A}\tilde{x}+\tilde{B}\tilde{v}$ with an uncertainty of ${\tilde{\Sigma }}^{w}$. The Gaussian distribution becomes:(20)$$p\left({\tilde{x}}^{\prime }/\tilde{x},\tilde{v},\vartheta ,\lambda \right)=\frac{1}{\sqrt{{\left(2\pi \right)}^{n(p+1)}\left|{\tilde{\Sigma }}^{w}\right|}}e^{-\frac{1}{2}{\left(D^{x}\tilde{x}-\tilde{A}\tilde{x}-\tilde{B}\tilde{v}\right)}^{T}{\tilde{\Pi }}^{w}\left(D^{x}\tilde{x}-\tilde{A}\tilde{x}-\tilde{B}\tilde{v}\right)}.$$

### 5.2. Prior Distributions

The remaining distributions $p\left(\tilde{v}\right),\;p\left(\theta \right)$ and p(λ) are the priors of the robot brain that can be transferred from prior (learned) experiences to the inference process. Similar to the previous section, a Gaussian prior is placed over the inputs $p\left(\tilde{v}\right)=N\left(\tilde{v}:\eta^{\tilde{v}},L^{\tilde{v}}\right)$ as well, with mean $\eta^{\tilde{v}}$ and prior covariance $L^{\tilde{v}}={\left(P^{\tilde{v}}\right)}^{-1}$, as:(21)$$p\left(\tilde{v}\right)=\frac{1}{\sqrt{{\left(2\pi \right)}^{r(d+1)}\left|L^{\tilde{v}}\right|}}e^{-\frac{1}{2}{\left(\tilde{v}-\eta^{\tilde{v}}\right)}^{T}P^{\tilde{v}}\left(\tilde{v}-\eta^{\tilde{v}}\right)}.$$The prior distribution of parameters $\theta \in R^{l}$ is assumed to be a Gaussian-centred around the prior parameter value $\eta^{\theta }$, with the prior covariance $L^{\theta }={\left(P^{\theta }\right)}^{-1}$:(22)$$p\left(\theta \right)=N\left(\theta :\eta^{\theta },L^{\theta }\right)=\frac{1}{\sqrt{{\left(2\pi \right)}^{l}\left|L^{\theta }\right|}}e^{-\frac{1}{2}{\left(\theta -\eta^{\theta }\right)}^{T}P^{\theta }\left(\theta -\eta^{\theta }\right)}.$$Similarly, a Gaussian prior is placed over the hyperparameters $\lambda \in R^{2}$:(23)$$p\left(\lambda \right)=N\left(\lambda :\eta^{\lambda },L^{\lambda }\right)=\frac{1}{\sqrt{{\left(2\pi \right)}^{2}\left|L^{\lambda }\right|}}e^{-\frac{1}{2}{\left(\lambda -\eta^{\lambda }\right)}^{T}P^{\lambda }\left(\lambda -\eta^{\lambda }\right)}.$$A higher value of $P^{\tilde{v}},\;P^{\theta }$ and $P^{\lambda }$ represents the robot’s higher confidence in its prior estimates $\eta^{\tilde{v}}$, $\eta^{\theta }$, and $\eta^{\lambda }$, respectively.

### 5.3. Simplification of the Internal Energy Action $\bar{U}$

This section aims at using the distributions from the previous sections to simplify $\bar{U}$. The logarithm of a Gaussian prior after dropping constants takes the general form:(24)$$\begin{array}{cc} \ln p(\theta )= & \ln N\left(\theta :\eta^{\theta },L^{\theta }\right)=-\frac{1}{2}{\left(\theta -\eta^{\theta }\right)}^{T}{\left(L^{\theta }\right)}^{-1}\left(\theta -\eta^{\theta }\right)-\frac{1}{2}\ln\left|L^{\theta }\right| \end{array}$$Therefore, substituting Equations (Equation 19)–(Equation 23) in Equation (Equation 18) and simplifying it using the prediction error terms $\epsilon^{\tilde{x}}=D^{x}\tilde{x}-\tilde{A}\tilde{x}-\tilde{B}\tilde{v},\;\epsilon^{\tilde{y}}=\tilde{y}-\tilde{C}\tilde{x},\;\epsilon^{\tilde{v}}=\tilde{v}-\eta^{\tilde{v}}\;\epsilon^{\theta }=\theta -\eta^{\theta },$ and $\epsilon^{\lambda }=\lambda -\eta^{\lambda }$, after dropping constants, yields:(25)$$\begin{array}{cc} U(\vartheta ,y)= & -\frac{1}{2}\epsilon^{\theta T}P^{\theta }\epsilon^{\theta }+\frac{1}{2}\ln|P^{\theta }|-\frac{1}{2}\epsilon^{\lambda T}P^{\lambda }\epsilon^{\lambda }+\frac{1}{2}\ln|P^{\lambda }|-\frac{1}{2}{\tilde{\epsilon }}^{T}\tilde{\Pi }\tilde{\epsilon }+\frac{1}{2}\ln\left|\tilde{\Pi }\right|, \end{array}$$
where $\tilde{\epsilon }=\left[\begin{array}{c} \epsilon^{\tilde{y}}\\ \epsilon^{\tilde{v}}\\ \epsilon^{\tilde{x}} \end{array}\right]=\left[\begin{array}{c} \tilde{y}-\tilde{C}\tilde{x}\\ \tilde{v}-\eta^{\tilde{v}}\\ D^{x}\tilde{x}-\tilde{A}\tilde{x}-\tilde{B}\tilde{v} \end{array}\right]$, and $\tilde{\Pi }=diag\left({\tilde{\Pi }}^{z},P^{\tilde{v}},{\tilde{\Pi }}^{w}\right)$. Here diag(.,.) is the block diagonal operator. Grouping the internal energy terms of the temporal and nontemporal parameters yields:(26)U(ϑ,y)=U(t)+U(θ)+U(λ),
where
(27)$$U\left(t\right)=-\frac{1}{2}{\tilde{\epsilon }}^{T}\tilde{\Pi }\tilde{\epsilon }+\frac{1}{2}\ln\left|\tilde{\Pi }\right|.$$Summing up the internal energy of all the temporal parameters over time yields the action of internal energy as follows:(28)$$\begin{array}{cc} \bar{U} & =U\left(\theta \right)+U\left(\lambda \right)+\sum_{t}U\left(t\right)\\ \; & =-\frac{1}{2}\epsilon^{\theta T}P^{\theta }\epsilon^{\theta }+\frac{1}{2}\ln|P^{\theta }|-\frac{1}{2}\epsilon^{\lambda T}P^{\lambda }\epsilon^{\lambda }+\frac{1}{2}\ln|P^{\lambda }|-\sum_{t}\frac{1}{2}{\tilde{\epsilon }}^{T}\tilde{\Pi }\tilde{\epsilon }+\frac{1}{2}\sum_{t}\ln\left|\tilde{\Pi }\right|. \end{array}$$It can be observed from Equation (Equation 28) that the robot’s priors $\left(\eta^{\tilde{v}},P^{\tilde{v}},\eta^{\theta },P^{\theta },\eta^{\lambda },P^{\lambda }\right)$ enter the free energy objective through the internal energy term. Intuitively, this can be seen as the direct influence of the robot’s prior beliefs on the inference process through the mismatches in the robot’s predictions. These weighed prediction errors drive the robot’s desire to maintain an equilibrium between its internal model and the generative process in the environment. A large mismatch between the robot’s predictions and the data results in a large prediction error, which gets precision-weighted and enters the free energy objective through its internal energy.

## 6. Mean-Field Approximation

The second widely used assumption for the simplification of free energy objectives is the factorization of parameters into independent subdensities for the recognition density [18], given by:(29)$$q\left(\vartheta \right)=\prod_{i}q\left(\theta^{i}\right)=q\left(\tilde{x}\right)q\left(\tilde{v}\right)q\left(\theta \right)q\left(\lambda \right).$$This approximation assumes the conditional independence between subdensities (states and parameters, for example). The subdensities are assumed to interact with each other only through the mean-field quantities. The strong biological plausibility of this approximation in terms of biological brain’s inference process is delineated in [27]. The main advantage of this approximation is the simplification of $\bar{V}$ in Equation (Equation 15). However, the mathematical proof for this simplification is missing from the DEM literature [18]. Therefore, this section aims to fill this gap by deriving these proofs by delineating all the intermediate assumptions.

### 6.1. Simplification of the Entropy Action $\bar{H}$

The entropy of the density in Equation (Equation 14), given by $H{\left(\vartheta \right)}_{q(\vartheta )}=-{\langle \ln q\left(\vartheta \right)\rangle }_{q(\vartheta )}$, can be simplified for all parameters as:(30)$$H{\left(\vartheta \right)}_{q(\vartheta )}=-\int \int \int \int q\left(\tilde{x},\tilde{v},\theta ,\lambda \right)\ln q(\tilde{x},\tilde{v},\theta ,\lambda )d\tilde{x}d\tilde{v}d\theta d\lambda$$Substituting the mean-field assumption given in Equation (Equation 29) in Equation (Equation 30) and using the property of normalized recognition densities $\int q\left(\theta^{i}\right)d\theta^{i}=1$ yields:(31)$$H{\left(\vartheta \right)}_{q(\vartheta )}=H\left(\theta \right)+H\left(\lambda \right)+H\left(t\right),$$
where $H\left(\theta \right)=-{\langle \ln q\left(\theta \right)\rangle }_{q(\theta )}$, $H\left(\lambda \right)=-{\langle \ln q\left(\lambda \right)\rangle }_{q(\lambda )}$ and $H\left(t\right)=-{\langle \ln q\left(\tilde{x}\right)\rangle }_{q(\tilde{x})}-$${\langle \ln q\left(\tilde{v}\right)\rangle }_{q(\tilde{v})}$. We place the Laplace approximation over the marginals of the recognition densities of all parameters as:(32)$$\begin{array}{cc} q(\theta )= & N\left(\theta :\mu^{\theta },\Sigma^{\theta }\right)=\frac{1}{\sqrt{{\left(2\pi \right)}^{l}\left|\Sigma^{\theta }\right|}}e^{-\frac{1}{2}{\left(\theta -\mu^{\theta }\right)}^{T}\Pi^{\theta }\left(\theta -\mu^{\theta }\right)}\\ q(\lambda )= & N\left(\lambda :\mu^{\lambda },\Sigma^{\lambda }\right)=\frac{1}{\sqrt{{\left(2\pi \right)}^{2}\left|\Sigma^{\lambda }\right|}}e^{-\frac{1}{2}{\left(\lambda -\mu^{\lambda }\right)}^{T}\Pi^{\lambda }\left(\lambda -\mu^{\lambda }\right)}\\ q(\tilde{x})= & N\left(\tilde{x}:\mu^{\tilde{x}},\Sigma^{\tilde{x}}\right)=\frac{1}{\sqrt{{\left(2\pi \right)}^{n(p+1)}\left|\Sigma^{\tilde{x}}\right|}}e^{-\frac{1}{2}{\left(\tilde{x}-\mu^{\tilde{x}}\right)}^{T}\Pi^{\tilde{x}}\left(\tilde{x}-\mu^{\tilde{x}}\right)}\\ q(\tilde{v})= & N\left(\tilde{v}:\mu^{\tilde{v}},\Sigma^{\tilde{v}}\right)=\frac{1}{\sqrt{{\left(2\pi \right)}^{r(d+1)}\left|\Sigma^{\tilde{v}}\right|}}e^{-\frac{1}{2}{\left(\tilde{v}-\mu^{\tilde{v}}\right)}^{T}\Pi^{\tilde{v}}\left(\tilde{v}-\mu^{\tilde{v}}\right)}. \end{array}$$The recognition densities given in Equation (Equation 32) might look similar to the priors distributions given in Equations (Equation 21)–(Equation 23), mainly due to the common Gaussian distribution. The prior distributions are centered around the prior mean and prior covariances, whereas the recognition density is centered around the mean $\mu^{i}$ and conditional covariance $\Sigma^{i}$ of the *i*-th parameter set.

The action of entropy can be calculated by substituting Equation (Equation 32) in Equation (Equation 31) and summing up the entropy terms of time dependent parameters with respect to time. Upon dropping the constant terms, it simplifies to:(33)$$\begin{array}{c} \bar{H}=H\left(\theta \right)+H\left(\lambda \right)+\sum_{t}H\left(t\right)=\frac{1}{2}\ln|\Sigma^{\theta }|+\frac{1}{2}\ln|\Sigma^{\lambda }|+\frac{1}{2}\sum_{t}\ln|\Sigma^{\tilde{x}}|+\frac{1}{2}\sum_{t}\ln\left|\Sigma^{\tilde{v}}\right|, \end{array}$$Equation (Equation 33) shows how the uncertainty in the robot’s inference directly enters the objective $\bar{F}$, through $\bar{H}$.

### 6.2. Mean-Field Terms

Given the simplified expressions for $\bar{U}$ and $\bar{H}$, the next step towards finding $\bar{F}$ in Equation (Equation 15) is to evaluate $\bar{V}$ given by:(34)$$\begin{array}{c} \bar{V}=\int {\langle U\left(y,\vartheta \right)\rangle }_{q(\vartheta )}dt \end{array}$$U(y,ϑ) can be simplified using the second-degree Taylor series expansion near the mean $\mu^{\vartheta }=\left\{\mu^{\tilde{x}},\mu^{\tilde{v}},\mu^{\theta },\mu^{\lambda }\right\}$ as:(35)$$\begin{array}{cc} U(y,\tilde{x},\tilde{v},\theta ,\lambda )= & U\left(y,\mu^{\vartheta }\right)+\sum_{i=1}^{4}U{\left(y,\mu^{\vartheta }\right)}_{\theta^{i}}\left(\theta^{i}-\mu^{i}\right)+\sum_{i=1}^{4}\sum_{j=1}^{4}{\left(\theta^{i}-\mu^{i}\right)}^{T}U{\left(y,\mu^{\vartheta }\right)}_{\theta^{i}\theta^{j}}\left(\theta^{j}-\mu^{j}\right), \end{array}$$
where we use the shorthand $U\left(y,\mu^{\vartheta }\right)=U\left(y,\vartheta \right){|}_{\vartheta =\mu^{\vartheta }}$ and $U{\left(y,\mu^{\vartheta }\right)}_{\vartheta }=U{\left(y,\vartheta \right)}_{\vartheta }{|}_{\vartheta =\mu^{\vartheta }}$. This approximation of the internal energy has nontrivial implications in terms of the biological brain’s decision-making process [21]. The second order approximation is justified because the Laplace and mean-field approximations entail an internal energy that is quadratic in $\tilde{x}$, $\tilde{v}$, and θ, as given in Equation (Equation 25), thereby reducing all its higher-order derivatives to zero. Moreover, the higher derivatives of $U(y,\mu^{\vartheta })$ with respect to λ, reduce to zero because of the assumptions made in Section 9. By differentiating Equation (Equation 25) at the mean $\mu^{i}$, $U{\left(y,\mu^{\vartheta }\right)}_{\theta^{i}}$ can be found to be all zeroes, which upon substitution in Equation (Equation 35) simplifies to:(36)$$U\left(y,\vartheta \right)=U\left(y,\mu^{\vartheta }\right)+\sum_{i=1}^{4}\sum_{j=1}^{4}{\left(\theta^{i}-\mu^{i}\right)}^{T}U{\left(y,\mu^{\vartheta }\right)}_{\theta^{i}\theta^{j}}\left(\theta^{j}-\mu^{j}\right).$$Substituting it in Equation (Equation 34), upon simplification yields:(37)$$\bar{V}=\bar{U}\left(y,\mu^{\vartheta }\right)+\int Wdt$$
where
$$\begin{array}{cc} W & =\frac{1}{2}\sum_{i,j=1}^{4}{\langle {\left(\theta^{i}-\mu^{i}\right)}^{T}U{\left(y,\mu^{\vartheta }\right)}_{\theta^{i}\theta^{j}}\left(\theta^{j}-\mu^{j}\right)\rangle }_{q\left(\vartheta^{i}\right)q\left(\vartheta^{j}\right)}\\ \; & =\frac{1}{2}\sum_{i,j=1}^{4}{\langle tr\left({\left(\theta^{i}-\mu^{i}\right)}^{T}U{\left(y,\mu^{\vartheta }\right)}_{\theta^{i}\theta^{j}}\left(\theta^{j}-\mu^{j}\right)\right)\rangle }_{q\left(\vartheta^{i}\right)q\left(\vartheta^{j}\right)}\\ \; & =\frac{1}{2}\sum_{i,j=1}^{4}{\langle tr\left(\left(\theta^{j}-\mu^{j}\right){\left(\theta^{i}-\mu^{i}\right)}^{T}U{\left(y,\mu^{\vartheta }\right)}_{\theta^{i}\theta^{j}}\right)\rangle }_{q\left(\vartheta^{i}\right)q\left(\vartheta^{j}\right)}\\ \; & =\frac{1}{2}\sum_{i,j=1}^{4}tr\left[{\langle \left(\theta^{j}-\mu^{j}\right){\left(\theta^{i}-\mu^{i}\right)}^{T}\rangle }_{q\left(\vartheta^{i}\right)q\left(\vartheta^{j}\right)}U{\left(y,\mu^{\vartheta }\right)}_{\theta^{i}\theta^{j}}\right]=\frac{1}{2}\sum_{i,j=1}^{4}tr\left[\Sigma^{ij}U{\left(y,\mu^{\vartheta }\right)}_{\theta^{i}\theta^{j}}\right] \end{array}$$Since the parameters $\vartheta^{i}$ are assumed to be independent of each other, the covariance between them drops to zero, resulting in:(38)$$\begin{array}{cc} \bar{V}= & \;\bar{U}\left(y,\mu^{\vartheta }\right)+\frac{1}{2}\int \sum_{i=1}^{4}tr[\Sigma^{i}U{\left(y,\mu^{\vartheta }\right)}_{\theta^{i}\theta^{i}}]dt\\ = & \;\bar{U}\left(y,\mu^{\vartheta }\right)+\int \left[W^{\tilde{x}}+W^{\tilde{v}}+W^{\theta }+W^{\lambda }\right]dt, \end{array}$$
where
(39)$$W^{\vartheta^{i}}=\frac{1}{2}tr\left[\Sigma^{i}U{\left(y,\mu^{\vartheta }\right)}_{\theta^{i}\theta^{i}}\right]$$
is defined as the mean-field term. Equation (Equation 38) shows that $\bar{V}$ is the sum of internal energy action and the mean field terms. The simplification of $\bar{V}$ is one of the main advantages of using mean-field approximation. However, this approximation can be relaxed to build Generalized Filtering [37,42], which is mainly relevant to nonlinear identification. It involves the modeling of parameters and hyperparameters in generalized coordinates (together with states) for online system identification. However, in this work, we take a simpler approach.

## 7. Simplified Free Energy Objectives

This section aims at simplifying the free energy objectives using the results from Section 5 and Section 6.

### 7.1. Simplification of Free Action

$\bar{F}$ is simplified by substituting Equations (Equation 28), (Equation 33) and (Equation 38) into Equation (Equation 15), yielding:(40)$$\begin{array}{cc} \bar{F}= & -\frac{1}{2}\left(\epsilon^{\theta T}P^{\theta }\epsilon^{\theta }\right){|}_{\vartheta =\mu^{\vartheta }}+\frac{1}{2}\ln|P^{\theta }|-\frac{1}{2}\left(\epsilon^{\lambda T}P^{\lambda }\epsilon^{\lambda }\right){|}_{\vartheta =\mu^{\vartheta }}+\frac{1}{2}\ln|P^{\lambda }|-\sum_{t}\frac{1}{2}\left({\tilde{\epsilon }}^{T}\tilde{\Pi }\tilde{\epsilon }\right){|}_{\vartheta =\mu^{\vartheta }}\\ & +\frac{1}{2}\sum_{t}\left(\ln\right|\tilde{\Pi }{\left|)\right|}_{\vartheta =\mu^{\vartheta }}+\frac{1}{2}\sum_{t}\left(\ln\right|\Sigma^{\tilde{x}}\left|+\ln\right|\Sigma^{\tilde{v}}\left|\right)+\frac{1}{2}\ln|\Sigma^{\theta }|+\frac{1}{2}\ln\left|\Sigma^{\lambda }\right|\\ & +\frac{1}{2}\sum_{t}tr[\Sigma^{\tilde{x}}U{\left(y,\mu^{\vartheta }\right)}_{\tilde{x}\tilde{x}}+\Sigma^{\tilde{v}}U{\left(y,\mu^{\vartheta }\right)}_{\tilde{v}\tilde{v}}+\Sigma^{\theta }U{\left(y,\mu^{\vartheta }\right)}_{\theta \theta }+\Sigma^{\lambda }U{\left(y,\mu^{\vartheta }\right)}_{\lambda \lambda }]. \end{array}$$We highlight three important terms in Equation (Equation 40): the combined prediction error of (generalized) outputs, inputs, and states $E=\frac{1}{2}\left({\tilde{\epsilon }}^{T}\tilde{\Pi }\tilde{\epsilon }\right){|}_{\vartheta =\mu^{\vartheta }}$, the log determinant of noise precision $G=\left(\ln|\tilde{\Pi }|\right){|}_{\vartheta =\mu^{\vartheta }}$, and the mean field term $W^{\vartheta^{i}}=\frac{1}{2}tr\left[\Sigma^{i}U{\left(y,\mu^{\vartheta }\right)}_{\theta^{i}\theta^{i}}\right]$. These terms will be used rigorously in the rest of the document.

### 7.2. Simplification of the Parameter Precisions

One of the main advantages of Laplace approximation is the simple evaluation of the covariance associated with the parameter estimation. This is achieved by setting the gradient of the free action with respect to individual parameter covariance as zero. The free action gradients with respect to covariance of paramaters $\vartheta^{i}$ is given by:(41)$$\begin{array}{cc} \frac{\partial \bar{F}}{\partial \Sigma^{i}}= & \frac{\partial }{\partial \Sigma^{i}}\left(\frac{1}{2}\ln|\Sigma^{i}\left|+\frac{1}{2}\sum_{t}tr\left(\Sigma^{i}U{\left(y,\mu^{\vartheta }\right)}_{\vartheta^{i}\vartheta^{i}}\right)\right)\right)=\frac{1}{2}\left({\left(\Sigma^{i}\right)}^{-1}+\bar{U}{\left(y,\mu^{\vartheta }\right)}_{\vartheta^{i}\vartheta^{i}}\right). \end{array}$$The optimal parameter covariance is the covariance that maximizes the free action with zero gradients. Assuming that the parameter covariances are time-invariant, and equating the gradients to zero yields $\Pi^{i}=-\bar{U}{\left(y,\mu^{\vartheta }\right)}_{\vartheta^{i}\vartheta^{i}}$, resulting in:(42)$$\begin{array}{c} \Pi^{\tilde{x}}=-U{\left(y,\mu^{\vartheta }\right)}_{\tilde{x}\tilde{x}},\;\Pi^{\tilde{v}}=-U{\left(y,\mu^{\vartheta }\right)}_{\tilde{v}\tilde{v}},\;\Pi^{\theta }=-\bar{U}{\left(y,\mu^{\vartheta }\right)}_{\theta \theta },\;\Pi^{\lambda }=-\bar{U}{\left(y,\mu^{\vartheta }\right)}_{\lambda \lambda } \end{array}$$From Equation (Equation 42), it is clear that the precision of parameters can be estimated just by evaluating the negative curvature of the internal energy at the conditional mode. These precision values denote the confidence of the estimator in the parameter estimate. Ideally, the parameter precision increases as the estimation process proceeds. Intuitively, the robot is more confident about its estimates (higher precision) when its predictions on the sensory data show the least variance.

### 7.3. Free Action at Optimal Precision

Substituting Equation (Equation 42) into Equation (Equation 39) at optimal precisions results in constant mean field terms. Therefore, the mean field terms in the free action given in Equation (Equation 40), reduces to a constant under the optimal precision given by Equation (Equation 42). Therefore, the free action at optimal precision for an LTI system reduces to:(43)$$\begin{array}{cc} \bar{F}= & -\frac{1}{2}\sum_{t}\left[\underset{\mathrm{prediction}\;\mathrm{error}\;\mathrm{of}\;\mathrm{outputs}}{\underset{︸}{{\left(\tilde{y}-\tilde{C}\tilde{x}\right)}^{T}{\tilde{\Pi }}^{z}\left(\tilde{y}-\tilde{C}\tilde{x}\right)\left|{}_{\vartheta =\mu^{\vartheta }}\right.}}+\underset{\mathrm{prediction}\;\mathrm{error}\;\mathrm{of}\;\mathrm{inputs}}{\underset{︸}{{\left(\tilde{v}-\eta^{\tilde{v}}\right)}^{T}P^{\tilde{v}}\left(\tilde{v}-\eta^{\tilde{v}}\right)\left|{}_{\vartheta =\mu^{\vartheta }}\right.}}\right]\\ & -\frac{1}{2}\underset{\mathrm{prediction}\;\mathrm{error}\;\mathrm{of}\;\mathrm{parameters}}{\underset{︸}{{\left(\theta -\eta^{\theta }\right)}^{T}P^{\theta }\left(\theta -\eta^{\theta }\right)\left|{}_{\vartheta =\mu^{\vartheta }}\right.}}-\frac{1}{2}\underset{\mathrm{prediction}\;\mathrm{error}\;\mathrm{of}\;\mathrm{hyperparameters}}{\underset{︸}{{\left(\lambda -\eta^{\lambda }\right)}^{T}P^{\lambda }\left(\lambda -\eta^{\lambda }\right)\left|{}_{\vartheta =\mu^{\vartheta }}\right.}}\\ & -\frac{1}{2}\sum_{t}\underset{\mathrm{prediction}\;\mathrm{error}\;\mathrm{of}\;\left(\mathrm{generalized}\right)\;\mathrm{states}}{\underset{︸}{{\left(D^{x}\tilde{x}-\tilde{A}\tilde{x}-\tilde{B}\tilde{v}\right)}^{T}{\tilde{\Pi }}^{w}\left(D^{x}\tilde{x}-\tilde{A}\tilde{x}-\tilde{B}\tilde{v}\right)\left|{}_{\vartheta =\mu^{\vartheta }}\right.}}+\underset{\mathrm{state}\;\mathrm{and}\;\mathrm{input}\;\mathrm{entropy}}{\underset{︸}{\frac{1}{2}n_{t}\ln\left|\Sigma^{X}\right|}}\\ & +\frac{1}{2}n^{t}\underset{\mathrm{noise}\;\mathrm{entropy}}{\underset{︸}{\left[\ln|{\tilde{\Pi }}^{z}\left|+\ln\right|P^{\tilde{v}}\left|+\ln\right|{\tilde{\Pi }}^{w}|\right]\left|{}_{\vartheta =\mu^{\vartheta }}\right.}}+\underset{\mathrm{parameter}\;\mathrm{entropy}}{\underset{︸}{\frac{1}{2}\ln\left|\Sigma^{\theta }P^{\theta }\right|}}+\underset{\mathrm{hyperparameter}\;\mathrm{entropy}}{\underset{︸}{\frac{1}{2}\ln\left|\Sigma^{\lambda }P^{\lambda }\right|}} \end{array}$$Note that the free action is not a function of the latent variables ($\vartheta =\{\tilde{x},\tilde{v},\theta ,\lambda \}$) but the sufficient statistics ($\mu^{\vartheta },\Sigma^{\vartheta }$) of the approximate posterior. For example, the weighted output prediction error term of $\bar{F}$ is ${\left(\tilde{y}-\tilde{C}\tilde{x}\right)}^{T}{\tilde{\Pi }}^{z}\left(\tilde{y}-\tilde{C}\tilde{x}\right)\left|{}_{\vartheta =\mu^{\vartheta }}\right.={\left(\tilde{y}-\mu^{\tilde{C}}\mu^{\tilde{x}}\right)}^{T}{\tilde{\Pi }}^{z}\left|{}_{\lambda^{z}=\mu^{\lambda^{z}}}\right.\left(\tilde{y}-\mu^{\tilde{C}}\mu^{\tilde{x}}\right)$, where $\left|{}_{\vartheta =\mu^{\vartheta }}\right.$ denotes the evaluation at $\vartheta =\mu^{\vartheta },\left(\tilde{x}=\mu^{\tilde{x}},\tilde{v}=\mu^{\tilde{v}},\theta =\mu^{\theta },\lambda =\mu^{\lambda }\right)$. We regroup the time-dependent components into one variable $X=\left[\begin{array}{c} \mu^{\tilde{x}}\\ \mu^{\tilde{v}} \end{array}\right]$, and use capital letters for the mean estimates of time-independent components ($\Theta =\mu^{\theta },\Lambda =\mu^{\lambda }$). Note that X,Θ, and Λ are part of the generative model and are the mean estimate of the components of plant dynamics.

### 7.4. Equivalence with the EM Algorithm

One of the most popular approaches to solve the Expectiation Maximization (EM) algorithm for state space models is to use the Maximum Likelihood Estimation (MLE), where the objective function is the log-likelihood of data, given by [43]:(44)$$\begin{array}{cc} \; & \ln L=-\frac{1}{2}\sum_{t}{\left(x_{t}-Ax_{t-1}-Bu_{t-1}\right)}^{T}\Pi^{w}\left(x_{t}-Ax_{t-1}-Bu_{t-1}\right)-\frac{n_{t}}{2}\ln\left|\Sigma^{w}\right|\\ \; & -\frac{n_{t}}{2}\ln|\Sigma^{z}|-\frac{1}{2}\ln\left|\Sigma^{x}\right|-\frac{1}{2}{\left(x_{0}-\mu \right)}^{T}\Pi^{x}\left(x_{0}-\mu \right)-\frac{1}{2}\sum_{t}{\left(y_{t}-Cx_{t}\right)}^{T}\Pi^{z}\left(y_{t}-Cx_{t}\right). \end{array}$$Comparing the objective functions of EM and DEM given by Equation (Equation 44) and Equation (Equation 43), respectively, DEM is equivalent to EM when:the mean field terms are neglected,the generalized motion is not considered, andthe robot’s priors on ϑ are not considered.Therefore, DEM can be considered as a generalized version of the EM algorithm with additional capabilities.

The free action at optimal precision (Equation (Equation 43)) is the sum of prediction errors (PE) and entropy of generalized states, parameters, noise, and hyperparameters and is independent of mean-field terms. Although the mean-field terms turns into a constant at optimal precision, their gradients do not. This property is leveraged in Section 8 to account for the uncertainties in the parameter estimation during state estimation and vice versa, through the gradient of mean-field terms.

## 8. Update Rules for Estimation

The Free Energy objectives (Section 4), together with the two simplifying approximations (Section 5 and Section 6), will be combined with an optimization procedure to form the ultimate DEM algorithm. The optimization procedure itself consists of the following two sets of update rules:a gradient ascent over its free action $\bar{F}$ for the time invariant parameters θ and λ,a gradient ascent over its free energy *F* for the time varying parameters *X*,where *F* and $\bar{F}$ are related through $F=\frac{\partial \bar{F}}{\partial t}$. The core idea is that the time varying parameters $\tilde{x}$ and $\tilde{v}$ can be estimated online from the robot’s instantaneous free energy, whereas the time-invariant parameters θ and λ can be estimated from its free action after observing a sequence of data. Accordingly, the update rules for both the gradient ascends are given by
(45)$$\frac{\partial \mu^{i}}{\partial a}=k^{i}\frac{\partial \bar{F}}{\partial \mu^{i}},$$
for the $a^{th}$ parameter update of the time invariant parameters and
(46)$$\frac{\partial \mu^{i}}{\partial t}=D\mu^{i}+k^{i}\frac{\partial F}{\partial \mu^{i}},$$
for the time=varying parameters, where $k^{i}$ is the learning rate. The presence of $D\mu^{i}$ term in Equation (Equation 46) differentiates the update rule from the general gradient ascent equation used in machine learning. This is to accommodate the boundary condition that when *F* is maximized, $\frac{\partial F}{\partial \mu^{i}}=0$ and ${\dot{\mu }}^{i}=D\mu^{i}$. In other words, when the free energy is maximized, the motion of the generalized states becomes their generalized motion [44]. However, the update equations for the time-invariant parameters, θ and λ, do not require the $D\mu^{i}$ term. Therefore, the update equations at $t^{th}$ time, $a^{th}$ parameter update step, and $b^{th}$ hyperparameter update step, after regrouping the (generalized) states and inputs as $X=\left[\begin{array}{c} \mu^{\tilde{x}}\\ \mu^{\tilde{v}} \end{array}\right]$, is given by:(47)$$\frac{\partial X}{\partial t}=DX+k^{X}\frac{\partial F}{\partial X},\;\;\frac{\partial \Theta }{\partial a}=k^{\Theta }\frac{\partial \bar{F}}{\partial \Theta },\;\;\frac{\partial \Lambda }{\partial b}=k^{\Lambda }\frac{\partial \bar{F}}{\partial \Lambda },$$
where $D=diag(D^{x},D^{v})$. Note that the gradient update rules are written not on the latent variables ($\tilde{x},\tilde{v},\theta$ and λ), but on their mean estimates (X,Θ and Λ). Since the update rules should be implemented in the discrete domain for robotics applications, Equation (Equation 47) is discretized under local linearization with the corresponding Jacobians as $J^{X}=D+k^{X}\frac{\partial^{2}F}{\partial X^{2}},\;J^{\Theta }=k^{\Theta }\frac{\partial^{2}\bar{F}}{\partial \Theta^{2}}$, and $J^{\Lambda }=k^{\Lambda }\frac{\partial^{2}\bar{F}}{\partial \Lambda^{2}}$. The (generalized) state and input update at time *t*, parameter update at step *a* and hyperparameter update at step *b* are given by:(48)$$\begin{array}{cc} X_{t+\Delta t} & =X_{t}+\left(e^{J^{X}\Delta t}-I\right){\left(J^{X}\right)}^{-1}\frac{\partial X}{\partial t},\\ \Theta_{a+1} & =\Theta_{a}+\left(e^{J^{\Theta }}-I\right){\left(J^{\Theta }\right)}^{-1}\frac{\partial \Theta }{\partial a},\\ \Lambda_{b+1} & =\Lambda_{b}+\left(e^{J^{\Lambda }}-I\right){\left(J^{\Lambda }\right)}^{-1}\frac{\partial \Lambda }{\partial b}. \end{array}$$Equation (Equation 48) shows that the update rules are dependent only on the gradients and curvatures of the free energy objectives.

### 8.1. The DEM Algorithm

The DEM algorithm is an iterative model inversion algorithm that uses Equations (Equation 42) and (Equation 48) to perform estimation on causal dynamic systems. It can be expressed using three steps:D step: (generalized) state and input estimation,E step: parameter estimation,M step: noise hyperparameter estimation,a nomenclature that is similar to the EM algorithm. Figure 2 shows an intuitive block diagram that demonstrates the inference process of DEM as a coupled dynamics between D, E, and M steps. The data from the environment and the robot brain’s prior distributions are used to infer the generative process. The pseudocode given in Algorithm 1 demonstrates how DEM performs estimation using only the gradient and curvatures of the free energy objectives. The next sections will focus on deriving the algebraic expressions for these quantities.

### 8.2. Updated Equations for Estimation

The free energy gradients in Equation (Equation 47) can be evaluated by differentiating Equation (Equation 40) with $\mu^{i}$. Substituting the resulting expression in Equation (Equation 47), upon simplification yields:(49)$$\begin{array}{cc} \dot{X} & =DX+k^{X}\left[-E_{X}+W_{X}^{X}+W_{X}^{\Theta }+W_{X}^{\Lambda }\right]\\ \frac{\partial \Theta }{\partial a} & =k^{\Theta }\left[-P^{\theta }\epsilon^{\theta }\left|{}_{\vartheta =\mu^{\vartheta }}\right.+\sum_{t}\left(-E_{\Theta }+W_{\Theta }^{X}+W_{\Theta }^{\Theta }+W_{\Theta }^{\Lambda }\right)\right]\\ \frac{\partial \Lambda }{\partial b} & =k^{\Lambda }\left[-P^{\lambda }\epsilon^{\lambda }\left|{}_{\vartheta =\mu^{\vartheta }}\right.+\sum_{t}\left(-E_{\Lambda }+W_{\Lambda }^{X}+W_{\Lambda }^{\Theta }+W_{\Lambda }^{\Lambda }+G_{\Lambda }\right)\right], \end{array}$$
where $E_{\mu^{i}}=\frac{\partial E}{\partial \mu^{i}}=\frac{1}{2}\frac{\partial \left({\tilde{\epsilon }}^{T}\tilde{\Pi }\tilde{\epsilon }\right)\left|{}_{\vartheta =\mu^{\vartheta }}\right.}{\partial \mu^{i}}$ is the gradient of the prediction error with respect to $\mu^{i}$, $W_{\mu^{j}}^{\mu^{i}}=\frac{\partial W^{\mu^{i}}}{\partial \mu^{j}}=\frac{1}{2}\frac{\partial }{\partial \mu^{j}}tr\left[\Sigma^{i}U{\left(y,\mu^{\vartheta }\right)}_{\mu^{i}\mu^{i}}\right]$ is the gradient of the mean field term of $\mu^{i}$ with respect to $\mu^{j}$, and $G_{\Lambda }=\frac{1}{2}\frac{\partial }{\partial \Lambda }\ln|\tilde{\Pi }|\left|{}_{\vartheta =\mu^{\vartheta }}\right.$ is the gradient of the log-determinant of the precision matrix with respect to Λ. From Equation (Equation 49), the Jacobians that are required for the updates given in Equation (Equation 48) can be evaluated as:(50)$$\begin{array}{cc} J^{X} & =D+k^{X}\left[-E_{XX}+W_{XX}^{X}+W_{XX}^{\Theta }+W_{XX}^{\Lambda }\right]\\ J^{\Theta } & =k^{\Theta }\left[-P^{\theta }+\sum_{t}\left(-E_{\Theta \Theta }+W_{\Theta \Theta }^{X}+W_{\Theta \Theta }^{\Theta }+W_{\Theta \Theta }^{\Lambda }\right)\right]\\ J^{\Lambda } & =k^{\Lambda }\left[-P^{\lambda }+\sum_{t}\left(-E_{\Lambda \Lambda }+W_{\Lambda \Lambda }^{X}+W_{\Lambda \Lambda }^{\Theta }+W_{\Lambda \Lambda }^{\Lambda }+G_{\Lambda \Lambda }\right)\right] \end{array}$$The Jacobians are employed in different loops corresponding to the D, E, and M steps in the Algorithm 1.
**Algorithm 1:** Dynamics Expectation Maximization
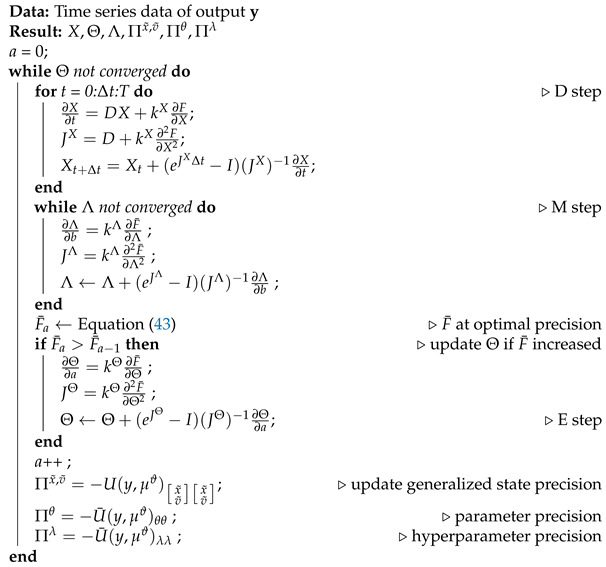


### 8.3. Update Equation for Precision of Estimates

The uncertainty in estimation is represented by the inverse of precision matrices $\Pi^{\vartheta^{i}}$. Differentiating Equations (Equation 25) and (Equation 28) twice and substituting it into Equation (Equation 42) upon simplification yield:(51)$$\begin{array}{cc} \Pi^{\tilde{x},\tilde{v}} & =-U{\left(y,\mu^{\vartheta }\right)}_{\left[\begin{array}{c} \tilde{x}\\ \tilde{v} \end{array}\right]\left[\begin{array}{c} \tilde{x}\\ \tilde{v} \end{array}\right]}=E_{XX}\\ \Pi^{\theta } & =-\bar{U}{\left(y,\mu^{\vartheta }\right)}_{\theta \theta }=P^{\theta }+\sum_{t}E_{\Theta \Theta }\\ \Pi^{\lambda } & =-\bar{U}{\left(y,\mu^{\vartheta }\right)}_{\lambda \lambda }=P^{\lambda }+\sum_{t}\left(E_{\Lambda \Lambda }-G_{\Lambda \Lambda })\right) \end{array}$$The only unknowns in the DEM update equations given by Equations (Equation 49)–(Equation 51) are the gradients and curvatures of *E*, *W*, and *G*. Section 9, Section 10 and Section 11 will deal with evaluating the simplified algebraic expressions for these gradients and curvatures. Using these simplifications, Section 12 will proceed towards expanding Algorithm 1.

## 9. Gradients of (Log Determinant of) Precision

This section aims at evaluating the gradients of the log determinant of noise precision ($G_{\Lambda },G_{\Lambda \Lambda }$) that are required for the hyperparameter update rules of the DEM algorithm. The precision matrix for hyperparameter estimation is modeled as:(52)$$\begin{array}{cc} \tilde{\Pi } & =diag\left({\tilde{\Pi }}^{z},P^{\tilde{v}},{\tilde{\Pi }}^{w}\right)=diag\left(e^{\lambda^{z}}\left(S⊗\Omega^{z}\right),P^{\tilde{v}},e^{\lambda^{w}}\left(S⊗\Omega^{w}\right)\right), \end{array}$$
where *S* is the noise smoothness matrix given by Equation (Equation 7) and $\Omega^{z},\Omega^{w}$ are the constant matrices given in Equation (Equation 6). Here the precision matrix is parametrized using $\lambda =\left[\begin{array}{c} \lambda^{z}\\ \lambda^{w} \end{array}\right]$, which is the only unknown in Equation (Equation 52). Therefore, the log determinant of precision and its gradients can be written as:(53)$$\begin{array}{cc} G & =\ln|\tilde{\Pi }|\left|{}_{\vartheta =\mu^{\vartheta }}\right.,\;G_{\Lambda }=\frac{1}{2}\left[\begin{array}{c} tr({\tilde{\Pi }}_{\lambda^{z}}{\tilde{\Pi }}^{-1})\\ tr({\tilde{\Pi }}_{\lambda^{w}}{\tilde{\Pi }}^{-1}) \end{array}\right]\left|{}_{\vartheta =\mu^{\vartheta }}\right.,\\ G_{\Lambda^{i}\Lambda^{j}} & =\frac{1}{2}tr\left({\tilde{\Pi }}_{\lambda^{i}\lambda^{j}}{\tilde{\Pi }}^{-1}-{\tilde{\Pi }}_{\lambda^{i}}{\tilde{\Pi }}^{-1}{\tilde{\Pi }}_{\lambda^{j}}{\tilde{\Pi }}^{-1}\right)\left|{}_{\vartheta =\mu^{\vartheta }}\right., \end{array}$$
where $\lambda^{i}$ is the $i^{th}$ element in λ. The gradients of precision in Equation (Equation 53) can be evaluated by differentiating Equation (Equation 52) as:(54)$$\begin{array}{cc} \; & {\tilde{\Pi }}_{\lambda^{z}}=diag\left(e^{\lambda^{z}}\left(S⊗\Omega^{z}\right),O,O\right)=diag\left({\tilde{\Pi }}^{z},O,O\right),\\ \; & {\tilde{\Pi }}_{\lambda^{w}}=diag\left(O,O,e^{\lambda^{w}}\left(S⊗\Omega^{z}\right)\right)=diag\left(O,O,{\tilde{\Pi }}^{z}\right),\\ \; & {\tilde{\Pi }}_{\lambda^{z}\lambda^{w}}=O,\;{\tilde{\Pi }}_{\lambda^{w}\lambda^{z}}=O. \end{array}$$Substituting Equations (Equation 52) and (Equation 54) in Equation (Equation 53) yields:(55)$$G_{\Lambda }=\frac{1}{2}\left[\begin{array}{c} tr(I_{n^{{\tilde{\Pi }}^{z}}})\\ tr(I_{n^{{\tilde{\Pi }}^{w}}}) \end{array}\right]=\frac{1}{2}\left[\begin{array}{c} n^{{\tilde{\Pi }}^{z}}\\ n^{{\tilde{\Pi }}^{w}} \end{array}\right],$$
where $I_{n^{{\tilde{\Pi }}^{z}}}$ is the identity matrix of size $n^{{\tilde{\Pi }}^{z}}$, which is the size of the ${\tilde{\Pi }}^{z}$ matrix. $\Pi^{w}$ and $\Pi^{z}$ are modeled to have an exponential relation with λ, so that any updates on λ would result in positive semi-definite precision matrices. However, this relation entails an infinitely differentiable precision matrix with respect to λ, increasing the computational complexity of the algorithm. Therefore, an approximation is made by forcefully setting ${\tilde{\Pi }}_{\lambda^{z}\lambda^{z}}={\tilde{\Pi }}_{\lambda^{w}\lambda^{w}}=O$, while maintaining the exponential relation between $\tilde{\Pi }$ and λ, thereby ensuring that the optimization process proceeds along the correct gradients ${\tilde{\Pi }}_{\lambda }$. Together with Equation (Equation 54), this approximation results in ${\tilde{\Pi }}_{\lambda \lambda }=O$. This assumption has two direct consequences:It simplifies all the update rules given in Equations (Equation 49) and (Equation 50),It simplifies the precision update rule for hyperparameters given in Equation (Equation 51).A direct consequence of this approximation is in the simplification of $G_{\Lambda \Lambda }$ in Equation (Equation 53), expressed as:(56)$$G_{\Lambda \Lambda }=-\frac{1}{2}\left[\begin{array}{cc} tr({\tilde{\Pi }}_{\lambda^{z}}{\tilde{\Pi }}^{-1}{\tilde{\Pi }}_{\lambda^{z}}{\tilde{\Pi }}^{-1}) & tr({\tilde{\Pi }}_{\lambda^{z}}{\tilde{\Pi }}^{-1}{\tilde{\Pi }}_{\lambda^{w}}{\tilde{\Pi }}^{-1})\\ tr({\tilde{\Pi }}_{\lambda^{w}}{\tilde{\Pi }}^{-1}{\tilde{\Pi }}_{\lambda^{z}}{\tilde{\Pi }}^{-1}) & tr({\tilde{\Pi }}_{\lambda^{w}}{\tilde{\Pi }}^{-1}{\tilde{\Pi }}_{\lambda^{w}}{\tilde{\Pi }}^{-1}) \end{array}\right]\left|{}_{\vartheta =\mu^{\vartheta }}\right.,$$
which upon substitution of Equations (Equation 52) and (Equation 54) yields:(57)$$G_{\Lambda \Lambda }=-\frac{1}{2}\left[\begin{array}{cc} tr(I_{n^{{\tilde{\Pi }}^{z}}}) & O\\ O & tr(I_{n^{{\tilde{\Pi }}^{w}}}) \end{array}\right]=-\frac{1}{2}\left[\begin{array}{cc} n^{{\tilde{\Pi }}^{z}} & O\\ O & n^{{\tilde{\Pi }}^{w}} \end{array}\right],$$
where $n^{{\tilde{\Pi }}^{z}}=m\left(p+1\right)$ and $n^{{\tilde{\Pi }}^{w}}=n\left(p+1\right)$ are the sizes of ${\tilde{\Pi }}^{z}$ and ${\tilde{\Pi }}^{w}$, respectively. From Equations (Equation 55) and (Equation 57), $G_{\Lambda }$ and $G_{\Lambda \Lambda }$ are constants, and can be pre-computed in Algorithm 1.

## 10. Gradients of Prediction Error

As opposed to the result above, the gradients of the prediction errors are *not* constant, as is shown in this section.

### 10.1. Gradients of Prediction Error along (Generalized) States

The error in prediction of (generalized) outputs, inputs and states is represented together by $\tilde{\epsilon }$, which makes up the precision weighted prediction error defined by $E=\frac{1}{2}{\tilde{\epsilon }}^{T}\tilde{\Pi }\tilde{\epsilon }\left|{}_{\vartheta =\mu^{\vartheta }}\right.$, where $\tilde{\Pi }=diag\left({\tilde{\Pi }}^{z},P^{\tilde{v}},{\tilde{\Pi }}^{w}\right)$. The error and its gradient are:(58)$$\tilde{\epsilon }=\left[\begin{array}{c} \tilde{y}-\tilde{C}\tilde{x}\\ \tilde{v}-\eta^{\tilde{v}}\\ D^{x}\tilde{x}-\tilde{A}\tilde{x}-\tilde{B}\tilde{v} \end{array}\right],{\tilde{\epsilon }}_{X}\left|{}_{\vartheta =\mu^{\vartheta }}\right.=\left[\begin{array}{cc} -\tilde{C} & O\\ O & I\\ D^{x}-\tilde{A} & -\tilde{B} \end{array}\right].$$The gradient of prediction error with respect to *X* can be simplified as:(59)$$E_{X}={\tilde{\epsilon }}_{X}^{T}\tilde{\Pi }\tilde{\epsilon }\left|{}_{\vartheta =\mu^{\vartheta }}\right.=A_{1}X+B_{1}\left[\begin{array}{c} \tilde{y}\\ -\eta^{\tilde{v}} \end{array}\right],$$
where $A_{1}=\left[\begin{array}{cc} {\tilde{C}}^{T}{\tilde{\Pi }}^{z}\tilde{C}+{\left(D^{x}-\tilde{A}\right)}^{T}{\tilde{\Pi }}^{w}\left(D^{x}-\tilde{A}\right) & -{\left(D^{x}-\tilde{A}\right)}^{T}{\tilde{\Pi }}^{w}\tilde{B}\\ -{\tilde{B}}^{T}{\tilde{\Pi }}^{w}\left(D^{x}-\tilde{A}\right) & P^{\tilde{v}}+{\tilde{B}}^{T}{\tilde{\Pi }}^{w}\tilde{B} \end{array}\right]\left|{}_{\vartheta =\mu^{\vartheta }}\right.$ and $B_{1}=\left[\begin{array}{cc} -{\tilde{C}}^{T}{\tilde{\Pi }}^{z} & O\\ O & P^{\tilde{v}} \end{array}\right]\left|{}_{\vartheta =\mu^{\vartheta }}\right.$. Since $E_{X}$ is linear in *X*, differentiating Equation (Equation 59) with respect to *X* yields a simple expression for curvature $E_{XX}$ as:(60)$$\begin{array}{c} E_{XX}={\tilde{\epsilon }}_{X}^{T}\tilde{\Pi }{\tilde{\epsilon }}_{X}\left|{}_{\vartheta =\mu^{\vartheta }}\right.=\left[\begin{array}{cc} {\tilde{C}}^{T}{\tilde{\Pi }}^{z}\tilde{C}+{\left(D^{x}-\tilde{A}\right)}^{T}{\tilde{\Pi }}^{w}\left(D^{x}-\tilde{A}\right) & -{\left(D^{x}-\tilde{A}\right)}^{T}{\tilde{\Pi }}^{w}\tilde{B}\\ -{\tilde{B}}^{T}{\tilde{\Pi }}^{w}\left(D^{x}-\tilde{A}\right) & P^{\tilde{v}}+{\tilde{B}}^{T}{\tilde{\Pi }}^{w}\tilde{B} \end{array}\right]\left|{}_{\vartheta =\mu^{\vartheta }}\right.. \end{array}$$

### 10.2. Gradients of Prediction Error along Parameters

To evaluate the gradients of prediction error along the parameters Θ, the reformulated definition of $\tilde{\epsilon }$ is used:(61)$$\tilde{\epsilon }=\left[\begin{array}{c} \tilde{y}-N\theta \\ \tilde{v}-\eta^{\tilde{v}}\\ D^{x}\tilde{x}-M\theta \end{array}\right],\;{\tilde{\epsilon }}_{\Theta }=\left({\tilde{\epsilon }}_{\theta }\right){|}_{\theta =\mu^{\theta }}=\left[\begin{array}{c} -N\\ O\\ -M \end{array}\right],$$
where *M* and *N* are given in Equation (Equation 5). This is to ensure that the variable Θ can be separated out of the expression for $E_{\Theta }$ such that it is linear in Θ as follows:(62)$$\begin{array}{cc} E_{\Theta } & =\left({\tilde{\epsilon }}_{\theta }^{T}\tilde{\Pi }\tilde{\epsilon }\right){|}_{\vartheta =\mu^{\vartheta }}=-\left[\begin{array}{ccc} N^{T} & O & M^{T} \end{array}\right]\tilde{\Pi }\left[\begin{array}{c} \tilde{y}-N\theta \\ \tilde{v}-\tilde{\eta }\\ D^{x}\tilde{x}-M\theta \end{array}\right]\left|{}_{\vartheta =\mu^{\vartheta }}\right.\\ \; & =\left(N^{T}{\tilde{\Pi }}^{z}N+M^{T}{\tilde{\Pi }}^{w}M\right)\left|{}_{\vartheta =\mu^{\vartheta }}\right.\Theta -\left[\begin{array}{cc} N^{T}{\tilde{\Pi }}^{z} & M^{T}{\tilde{\Pi }}^{w}D^{x} \end{array}\right]\left|{}_{\vartheta =\mu^{\vartheta }}\right.\left[\begin{array}{c} \tilde{y}\\ \mu^{\tilde{x}} \end{array}\right]\\ \; & =A_{2}\Theta -B_{2}\left[\begin{array}{c} \tilde{y}\\ \mu^{\tilde{x}} \end{array}\right]. \end{array}$$Since $E_{\Theta }$ is linear in Θ, differentiating Equation (Equation 62) with respect to Θ yields a simple expression for $E_{\Theta \Theta }$ as:(63)$$E_{\Theta \Theta }=\left({\tilde{\epsilon }}_{\theta }^{T}\tilde{\Pi }{\tilde{\epsilon }}_{\theta }\right)\left|{}_{\vartheta =\mu^{\vartheta }}\right.=\left[N^{T}{\tilde{\Pi }}^{z}N+M^{T}{\tilde{\Pi }}^{w}M\right]\left|{}_{\vartheta =\mu^{\vartheta }}\right.=A_{2}.$$

### 10.3. Gradients of Prediction Error along Hyperparameters

The gradients of prediction error along the hyperparameters Λ is simpler, and is given by $E_{\Lambda }=\frac{1}{2}{\tilde{\epsilon }}^{T}{\tilde{\Pi }}_{\lambda }\tilde{\epsilon }\left|{}_{\vartheta =\mu^{\vartheta }}\right.,\;E_{\Lambda \Lambda }=\frac{1}{2}{\tilde{\epsilon }}^{T}{\tilde{\Pi }}_{\lambda \lambda }\tilde{\epsilon }\left|{}_{\vartheta =\mu^{\vartheta }}\right.$, which upon using ${\tilde{\Pi }}_{\lambda \lambda }=0$, gives:(64)$$\begin{array}{cc} E_{\Lambda } & =\frac{1}{2}tr\left({\tilde{\Pi }}_{\lambda }\tilde{\epsilon }{\tilde{\epsilon }}^{T}\right)\left|{}_{\vartheta =\mu^{\vartheta }}\right.=\frac{1}{2}\left[\begin{array}{c} tr({\tilde{\Pi }}^{z}\left(\tilde{y}-\tilde{C}\tilde{x}\right){\left(\tilde{y}-\tilde{C}\tilde{x}\right)}^{T})\\ tr({\tilde{\Pi }}^{w}\left(D^{x}\tilde{x}-\tilde{A}\tilde{x}-\tilde{B}\tilde{u}\right){\left(D^{x}\tilde{x}-\tilde{A}\tilde{x}-\tilde{B}\tilde{u}\right)}^{T}) \end{array}\right]\left|{}_{\vartheta =\mu^{\vartheta }}\right.,\\ E_{\Lambda \Lambda } & =\frac{1}{2}{\tilde{\epsilon }}^{T}{\tilde{\Pi }}_{\lambda \lambda }\tilde{\epsilon }\left|{}_{\vartheta =\mu^{\vartheta }}\right.=0. \end{array}$$In summary, Equations (Equation 59), (Equation 62) and (Equation 64) represent the gradients of the prediction error term, whereas Equations (Equation 60), (Equation 63) and (Equation 64) represent its curvatures. The next section will deal with evaluating the analytic expressions for all the gradients and curvatures of the mean field term.

## 11. Gradients of Mean Field Terms

This section aims to derive the analytic expressions for the mean field terms and their gradients: $W_{\mu^{j}}^{\mu^{i}}$ and $W_{\mu^{j}\mu^{j}}^{\mu^{i}},\;\forall \;\mu \in \left\{X,\Theta ,\Lambda \right\}$.

### 11.1. Gradients of Mean Field Terms along Hyperparameters

In this section, we prove that all the gradients and curvatures of $W^{\Lambda }$ (namely $W_{\mu^{i}}^{\Lambda }$ and $W_{\mu^{i}\mu^{i}}^{\Lambda }$) are zeroes. The mean field term for hyperparameters Λ can be expressed as:(65)$$W^{\Lambda }=\frac{1}{2}tr\left[\Sigma^{\lambda }U{\left(y,\mu^{\vartheta }\right)}_{\lambda \lambda }\right].$$To compute the gradients of $W^{\Lambda }$, we need the curvature of internal energy with respect to λ. This can be evaluated by first differentiating Equation (Equation 25) with respect to λ and evaluating it at $\vartheta =\mu^{\vartheta }$, which yields:(66)$$\begin{array}{cc} U{\left(y,\mu^{\vartheta }\right)}_{\lambda }=U{\left(y,\vartheta \right)}_{\lambda }\left|{}_{\vartheta =\mu^{\vartheta }}\right. & =-P^{\lambda }\epsilon^{\lambda }\left|{}_{\vartheta =\mu^{\vartheta }}\right.-\frac{1}{2}\left({\tilde{\epsilon }}^{T}{\tilde{\Pi }}_{\lambda }\tilde{\epsilon }\right)\left|{}_{\vartheta =\mu^{\vartheta }}\right.+G_{\Lambda }, \end{array}$$
where $G_{\Lambda }$ is given by Equation (Equation 55). Upon further differentiation, we get:(67)$$\begin{array}{cc} U{\left(y,\mu^{\vartheta }\right)}_{\lambda \lambda } & =-P^{\lambda }-\frac{1}{2}\left({\tilde{\epsilon }}^{T}{\tilde{\Pi }}_{\lambda \lambda }\tilde{\epsilon }\right)\left|{}_{\vartheta =\mu^{\vartheta }}\right.+G_{\Lambda \Lambda }. \end{array}$$The assumption of ${\tilde{\Pi }}_{\lambda \lambda }=0$ applied to Equation (Equation 67) yields:(68)$$\begin{array}{c} U{\left(y,\mu^{\vartheta }\right)}_{\lambda \lambda }=-P^{\lambda }+G_{\Lambda \Lambda } \end{array}$$
which contains only constants. Therefore, the assumption of ${\tilde{\Pi }}_{\lambda \lambda }=0$ reduces all the gradients and curvatures of mean field terms of Λ to zeros:(69)$$W_{\mu^{i}}^{\Lambda }=\frac{1}{2}tr\left[\Sigma^{\lambda }U{\left(y,\mu^{\vartheta }\right)}_{\lambda \lambda \mu^{i}}\right]=0,\;W_{\mu^{i}\mu^{i}}^{\Lambda }=0.$$Since the internal energy given in Equation (Equation 25) is quadratic in $\vartheta^{i}$, and since ${\tilde{\Pi }}_{\lambda \lambda }=0$, all the gradients and curvatures of the mean field term of $\vartheta^{i}$ with respect to itself are zeros:(70)$$W_{\mu^{i}}^{\mu^{i}}=\frac{1}{2}tr\left[\Sigma^{\vartheta^{i}}U{\left(y,\mu^{\vartheta }\right)}_{\vartheta^{i}\vartheta^{i}\mu^{i}}\right]=0,\;W_{\mu^{i}\mu^{i}}^{\mu^{i}}=0.$$

### 11.2. Gradients of Mean Field Terms along Generalized States

The mean-field term of the combined generalized states *X* can be expressed as:(71)$$W^{X}=\frac{1}{2}tr\left[\Sigma^{X}U{\left(y,\mu^{\vartheta }\right)}_{XX}\right].$$The curvature of internal energy with respect to *X* can be calculated by differentiating Equation (Equation 25) with respect to *X* twice, resulting in $U{\left(y,\mu^{\vartheta }\right)}_{XX}=-\frac{1}{2}{\tilde{\epsilon }}_{X}^{T}\tilde{\Pi }{\tilde{\epsilon }}_{X}$. Substituting it in Equation (Equation 71) upon differentiation with Θ yields:(72)$$\begin{array}{cc} W_{\Theta^{i}}^{X} & =-\frac{1}{2}tr\left(\Sigma^{X}{\tilde{\epsilon }}_{X\theta^{i}}^{T}\tilde{\Pi }{\tilde{\epsilon }}_{X}\right)\left|{}_{\vartheta =\mu^{\vartheta }}\right.=-\frac{1}{2}tr\left(\Sigma^{X}\left[\begin{array}{ccc} -{\tilde{C}}_{\theta^{i}}^{T} & O & -{\tilde{A}}_{\theta^{i}}^{T}\\ O & O & -{\tilde{B}}_{\theta^{i}}^{T} \end{array}\right]\tilde{\Pi }\left[\begin{array}{cc} -\tilde{C} & O\\ O & I\\ D^{x}-\tilde{A} & -\tilde{B} \end{array}\right]\right)\left|{}_{\vartheta =\mu^{\vartheta }}\right.\\ & =-\frac{1}{2}tr\left[\Sigma^{X}\left[\begin{array}{cc} {\tilde{C}}_{\theta^{i}}^{T}{\tilde{\Pi }}^{z}\tilde{C}-{\tilde{A}}_{\theta^{i}}^{T}{\tilde{\Pi }}^{w}\left(D^{x}-\tilde{A}\right) & {\tilde{A}}_{\theta^{i}}^{T}{\tilde{\Pi }}^{w}\tilde{B}\\ -{\tilde{B}}_{\theta^{i}}^{T}{\tilde{\Pi }}^{w}\left(D^{x}-\tilde{A}\right) & {\tilde{B}}_{\theta^{i}}^{T}{\tilde{\Pi }}^{w}\tilde{B} \end{array}\right]\right]\left|{}_{\vartheta =\mu^{\vartheta }}\right., \end{array}$$
where the gradient of the mean field with respect to Θ is given by $W_{\Theta }^{X}={\left[\begin{array}{cccc} W_{\Theta^{1}}^{X} & W_{\Theta^{2}}^{X} & \ldots & W_{\Theta^{i}}^{X} \end{array}\right]}^{T}$. Similarly, the elements of the curvature matrix of the mean field term with respect to Θ is given by:(73)$$\begin{array}{c} W_{\Theta^{i}\Theta^{j}}^{X}=-\frac{1}{2}tr\left(\Sigma^{X}{\tilde{\epsilon }}_{X\theta^{i}}^{T}\tilde{\Pi }{\tilde{\epsilon }}_{X\theta^{j}}\right)\left|{}_{\vartheta =\mu^{\vartheta }}\right.=-\frac{1}{2}tr\left[\Sigma^{X}\left[\begin{array}{cc} {\tilde{C}}_{\theta^{i}}^{T}{\tilde{\Pi }}^{z}{\tilde{C}}_{\theta^{j}}+{\tilde{A}}_{\theta^{i}}^{T}{\tilde{\Pi }}^{w}{\tilde{A}}_{\theta^{j}} & {\tilde{A}}_{\theta^{i}}^{T}{\tilde{\Pi }}^{w}{\tilde{B}}_{\theta^{j}}\\ {\tilde{B}}_{\theta^{i}}^{T}{\tilde{\Pi }}^{w}{\tilde{A}}_{\theta^{j}} & {\tilde{B}}_{\theta^{i}}^{T}{\tilde{\Pi }}^{w}{\tilde{B}}_{\theta^{j}}^{T} \end{array}\right]\right]\left|{}_{\vartheta =\mu^{\vartheta }}\right.. \end{array}$$The gradient and curvature of the mean field term of *X* with respect to Λ can be evaluated as:(74)$$\begin{array}{cc} W_{\Lambda }^{X} & =-\frac{1}{2}tr\left(\Sigma^{X}{\tilde{\epsilon }}_{X}^{T}{\tilde{\Pi }}_{\lambda }{\tilde{\epsilon }}_{X}\right)\left|{}_{\vartheta =\mu^{\vartheta }}\right.=-\frac{1}{2}tr\left({\tilde{\Pi }}_{\lambda }{\tilde{\epsilon }}_{X}\Sigma^{X}{\tilde{\epsilon }}_{X}^{T}\right)\left|{}_{\vartheta =\mu^{\vartheta }}\right.\\ & =-\frac{1}{2}\left[\begin{array}{c} tr({\tilde{\Pi }}^{z}\tilde{C}\Sigma^{x}{\tilde{C}}^{T})\\ tr({\tilde{\Pi }}^{w}\left[\begin{array}{cc} (D^{x}-\tilde{A}), & -\tilde{B} \end{array}\right]\Sigma^{X}\left[\begin{array}{c} {\left(D^{x}-\tilde{A}\right)}^{T}\\ -{\tilde{B}}^{T} \end{array}\right]) \end{array}\right]\left|{}_{\vartheta =\mu^{\vartheta }}\right.\\ W_{\Lambda \Lambda }^{X} & =-\frac{1}{2}tr\left({\tilde{\Pi }}_{\lambda \lambda }{\tilde{\epsilon }}_{X}\Sigma^{X}{\tilde{\epsilon }}_{X}^{T}\right)\left|{}_{\vartheta =\mu^{\vartheta }}\right.=0, \end{array}$$
where $\Sigma^{x}$ is a component of $\Sigma^{X}=[\begin{array}{cc} \Sigma^{\tilde{x}} & \Sigma^{\tilde{x}\tilde{v}}\\ \Sigma^{\tilde{x}\tilde{v}} & \Sigma^{\tilde{v}} \end{array}]$. Here the curvature $W_{\Lambda \Lambda }^{X}$ vanishes due to the assumption that ${\tilde{\Pi }}_{\lambda \lambda }=0$.

### 11.3. Gradients of Mean Field Terms along Parameters

The mean-field term of the parameters Θ can be expressed as
(75)$$W^{\Theta }=\frac{1}{2}tr\left[\Sigma^{\theta }U{\left(y,\mu^{\vartheta }\right)}_{\theta \theta }\right]=\frac{1}{2}tr\left[\Sigma^{\theta }\left(-P^{\theta }-{\tilde{\epsilon }}_{\theta }^{T}\tilde{\Pi }{\tilde{\epsilon }}_{\theta }\right)\right]\left|{}_{\vartheta =\mu^{\vartheta }}\right..$$Differentiating Equation (Equation 75) with *X* and substituting Equation (Equation 61) in it yields the gradient as:(76)$$W_{X}^{\Theta }=-\frac{1}{2}\left[\begin{array}{c} tr\left(\Sigma^{\theta }\left(N_{X^{1}}^{T}{\tilde{\Pi }}^{z}N+M_{X^{1}}^{T}{\tilde{\Pi }}^{w}M\right)\right)\\ tr\left(\Sigma^{\theta }\left(N_{X^{2}}^{T}{\tilde{\Pi }}^{z}N+M_{X^{2}}^{T}{\tilde{\Pi }}^{w}M\right)\right)\\ \ldots \end{array}\right]\left|{}_{\vartheta =\mu^{\vartheta }}\right.,$$
and the elements of the curvature matrix as:(77)$$W_{X^{i}X^{j}}^{\Theta }=-\frac{1}{2}tr\left(\Sigma^{\theta }\left(N_{X^{i}}^{T}{\tilde{\Pi }}^{z}N_{X^{j}}+M_{X^{i}}^{T}{\tilde{\Pi }}^{w}M_{X^{j}}^{T}\right)\right)\left|{}_{\vartheta =\mu^{\vartheta }}\right..$$Differentiating Equation (Equation 75) with respect to λ yields the gradient and curvature as:(78)$$\begin{array}{cc} W_{\Lambda }^{\Theta } & =-\frac{1}{2}tr\left(\Sigma^{\theta }{\tilde{\epsilon }}_{\theta }^{T}{\tilde{\Pi }}_{\lambda }{\tilde{\epsilon }}_{\theta }\right)\left|{}_{\vartheta =\mu^{\vartheta }}\right.=-\frac{1}{2}tr\left({\tilde{\Pi }}_{\lambda }{\tilde{\epsilon }}_{\theta }\Sigma^{\theta }{\tilde{\epsilon }}_{\theta }^{T}\right)\left|{}_{\vartheta =\mu^{\vartheta }}\right.,\\ \; & =-\frac{1}{2}\left[\begin{array}{c} tr({\tilde{\Pi }}_{\lambda^{z}}{\tilde{\epsilon }}_{\theta }\Sigma^{\theta }{\tilde{\epsilon }}_{\theta }^{T})\\ tr({\tilde{\Pi }}_{\lambda^{w}}{\tilde{\epsilon }}_{\theta }\Sigma^{\theta }{\tilde{\epsilon }}_{\theta }^{T}) \end{array}\right]\left|{}_{\vartheta =\mu^{\vartheta }}\right.=-\frac{1}{2}\left[\begin{array}{c} tr({\tilde{\Pi }}^{z}N\Sigma^{\theta }N^{T})\\ tr({\tilde{\Pi }}^{w}M\Sigma^{\theta }M^{T}) \end{array}\right]\left|{}_{\vartheta =\mu^{\vartheta }}\right.\\ W_{\Lambda \Lambda }^{\Theta } & =-\frac{1}{2}tr\left({\tilde{\Pi }}_{\lambda \lambda }{\tilde{\epsilon }}_{\theta }\Sigma^{\theta }{\tilde{\epsilon }}_{\theta }^{T}\right)\left|{}_{\vartheta =\mu^{\vartheta }}\right.=0. \end{array}$$Here, $W_{\Lambda \Lambda }^{\Theta }$ vanishes due to the assumption that ${\tilde{\Pi }}_{\lambda \lambda }=0$.

## 12. The Complete DEM Algorithm

By combining the gradients found from Section 9, Section 10, and Section 11 with the Algorithm 1, we can finalize the full DEM algorithm so that it can iteratively compute the estimates and the associated precisions from data.

### 12.1. DEM Estimates

The main equations that are required to perform the update rules of DEM given in Equation (Equation 48) can be summarized as:(79)$$\begin{array}{cc} \dot{X} & =DX+k^{X}\left(-E_{X}+W_{X}^{\Theta }\right),\;J^{X}=D+k^{X}\left(-E_{XX}+W_{XX}^{\Theta }\right)\\ \frac{\partial \Theta }{\partial a} & =k^{\Theta }\left[-P^{\theta }\epsilon^{\theta }\left|{}_{\vartheta =\mu^{\vartheta }}\right.+\sum_{t}\left(-E_{\Theta }+W_{\Theta }^{X}\right)\right],\;J^{\Theta }=k^{\Theta }\left[-P^{\theta }+\sum_{t}\left(-E_{\Theta \Theta }+W_{\Theta \Theta }^{X}\right)\right]\\ \frac{\partial \Lambda }{\partial b} & =k^{\Lambda }\left[-P^{\lambda }\epsilon^{\lambda }\left|{}_{\vartheta =\mu^{\vartheta }}\right.+\sum_{t}\left(-E_{\Lambda }+W_{\Lambda }^{X}+W_{\Lambda }^{\Theta }+G_{\Lambda }\right)\right],\;J^{\Lambda }=k^{\Lambda }\left(-P^{\lambda }+n^{t}G_{\Lambda \Lambda }\right) \end{array}$$
where $E_{X},E_{XX},E_{\Theta },E_{\Theta \Theta },E_{\Lambda },W_{X}^{\Theta },W_{XX}^{\Theta },W_{\Theta }^{X},W_{\Theta \Theta }^{X},W_{\Lambda }^{X},$$W_{\Lambda }^{\Theta },G_{\Lambda },andG_{\Lambda \Lambda }$ are given by Equations (Equation 55), (Equation 57), (Equation 59), (Equation 60), (Equation 62)–(Equation 64), (Equation 72)–(Equation 74), (Equation 76)–(Equation 78), respectively. The hyperparameter update rule can be further simplified to reduce the computational complexity as:(80)$$\frac{\partial \Lambda }{\partial b}=-k^{\Lambda }P^{\lambda }\epsilon^{\lambda }\left|{}_{\vartheta =\mu^{\vartheta }}\right.+\frac{k^{\Lambda }n^{t}}{2}\left[\begin{array}{c} n^{{\tilde{\Pi }}^{z}}\\ n^{{\tilde{\Pi }}^{w}} \end{array}\right]-\frac{k^{\Lambda }}{2}\left[\begin{array}{c} tr({\tilde{\Pi }}^{z}A_{3})\\ tr({\tilde{\Pi }}^{w}B_{3}) \end{array}\right]\left|{}_{\vartheta =\mu^{\vartheta }}\right.\;$$
where
$$\begin{array}{cc} A_{3} & =\sum_{t}\left(\left(\tilde{y}-\tilde{C}\tilde{x}\right){\left(\tilde{y}-\tilde{C}\tilde{x}\right)}^{T}+N\Sigma^{\theta }N^{T}+\tilde{C}\Sigma^{\tilde{x}\tilde{x}}{\tilde{C}}^{T}\right)\\ B_{3} & =\sum_{t}\left(\left(D^{x}\tilde{x}-\tilde{A}\tilde{x}-\tilde{B}\tilde{u}\right){\left(D^{x}\tilde{x}-\tilde{A}\tilde{x}-\tilde{B}\tilde{u}\right)}^{T}+M\Sigma^{\theta }M^{T}+\left[\begin{array}{cc} (D^{x}-\tilde{A}), & -\tilde{B} \end{array}\right]\Sigma^{X}\left[\begin{array}{c} {\left(D^{x}-\tilde{A}\right)}^{T}\\ -{\tilde{B}}^{T} \end{array}\right]\right). \end{array}$$Substituting Equation (Equation 57) to the expression for $J^{\Lambda }$ in Equation (Equation 79) yields:(81)$$J^{\Lambda }=-k^{\Lambda }\left(P^{\lambda }+\frac{n^{t}}{2}\left[\begin{array}{cc} n^{{\tilde{\Pi }}^{z}} & O\\ O & n^{{\tilde{\Pi }}^{w}} \end{array}\right]\right),$$
which is independent of Λ. This reduces the algorithm’s computational complexity, as $J^{\Lambda }$ can now be pre-computed.

### 12.2. Precision of Estimates

This section simplifies the precision for DEM’s estimates for an LTI system. The confidence in the estimate of (generalized) states and inputs can be simplified using Equations (Equation 51) and (Equation 60) as:(82)$$\begin{array}{cc} \; & \Pi^{\tilde{x},\tilde{v}}=E_{XX}=\left[\begin{array}{cc} {\tilde{C}}^{T}{\tilde{\Pi }}^{z}\tilde{C}+{\left(D^{x}-\tilde{A}\right)}^{T}{\tilde{\Pi }}^{w}\left(D^{x}-\tilde{A}\right) & -{\left(D^{x}-\tilde{A}\right)}^{T}{\tilde{\Pi }}^{w}\tilde{B}\\ -{\tilde{B}}^{T}{\tilde{\Pi }}^{w}\left(D^{x}-\tilde{A}\right) & P^{\tilde{v}}+{\tilde{B}}^{T}{\tilde{\Pi }}^{w}\tilde{B} \end{array}\right]\left|{}_{\vartheta =\mu^{\vartheta }}\right.. \end{array}$$From Equation (Equation 82), the precisions for state and input estimation are $\Pi^{\tilde{x}\tilde{x}}={\tilde{C}}^{T}{\tilde{\Pi }}^{z}\tilde{C}+{\left(D^{x}-\tilde{A}\right)}^{T}{\tilde{\Pi }}^{w}\left(D^{x}-\tilde{A}\right)$ and $\Pi^{\tilde{v}\tilde{v}}=P^{\tilde{v}}+{\tilde{B}}^{T}{\tilde{\Pi }}^{w}\tilde{B}$, respectively. The cross-correlation between the (generalized) states and inputs are given by $-{\tilde{B}}^{T}{\tilde{\Pi }}^{w}\left(D^{x}-\tilde{A}\right)$. Since $\Pi^{\tilde{x},\tilde{v}}$ is independent of *X*, it can be updated outside the D step.

Combining the results of Equations (Equation 51) and (Equation 63) yields the precision of parameter estimates $\Pi^{\theta }$, which is independent of Θ, as:(83)$$\Pi^{\theta }=P^{\theta }+\sum_{t}E_{\Theta \Theta }=P^{\theta }+\sum_{t}\left[N^{T}{\tilde{\Pi }}^{z}N+M^{T}{\tilde{\Pi }}^{w}M\right]\left|{}_{\vartheta =\mu^{\vartheta }}\right..\;$$From Equations (Equation 51), (Equation 57) and (Equation 64), the precision of hyperparameter estimation is:(84)$$\Pi^{\lambda }=P^{\lambda }+\sum_{t}\left(E_{\Lambda \Lambda }-G_{\Lambda \Lambda })\right)=P^{\lambda }+\frac{n^{t}}{2}diag\left(n^{{\tilde{\Pi }}^{z}},n^{{\tilde{\Pi }}^{w}}\right),$$
which is a constant and hence is never updated in the algorithm. In conclusion, the estimation using Equation (Equation 79), along with the precision of these estimates given by Equations (Equation 82)–(Equation 84) completely define the DEM algorithm for an LTI system with colored noises. The complete DEM algorithm is given in Algorithm 2.
**Algorithm 2:** Dynamics Expectation Maximization
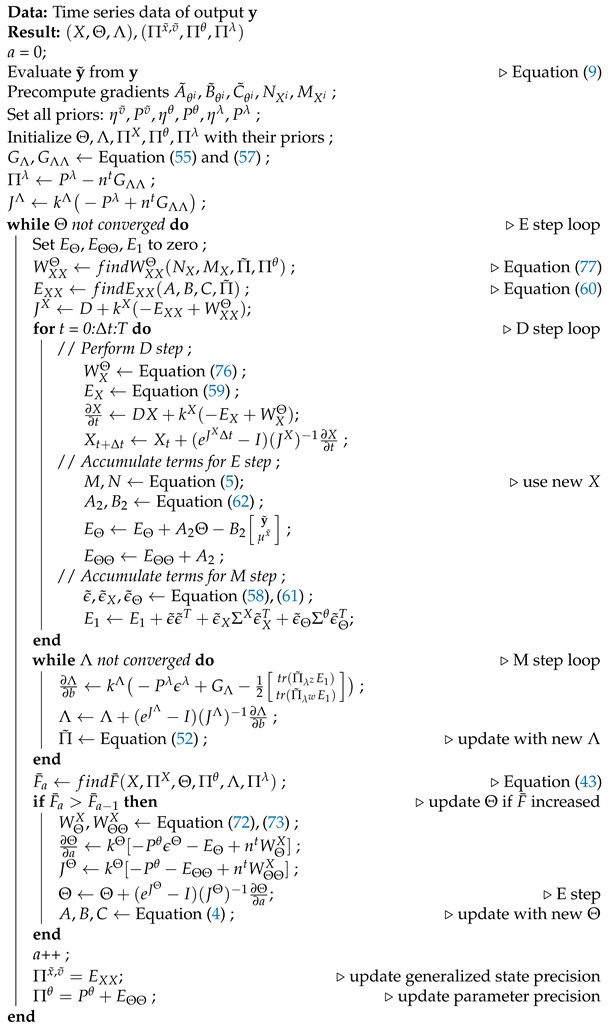


## 13. Translation into Simplified Mathematical Form

Although the pseudocode derived in the previous sections is sufficient to replicate the DEM algorithm for an LTI system with colored noise, it is not sufficient to analyze DEM using the standard control systems tools for stability checks, convergence, etc. Therefore, in this section we translate the algorithm into a simplified mathematical form that control engineers can easily analyze. The following subsections aim at converting the DEM updates into a coupled linear system.

### 13.1. State and Input Estimation as a Linear Observer

This section deals with reformulating the D step of DEM for an LTI system as a (generalized) state and input observer. Substituting Equation (Equation 59) in Equation (Equation 79) with a learning rate of $k^{X}=1$ yields [2]:(85)$$\dot{X}=\left[\begin{array}{c} \dot{\tilde{x}}\\ \dot{\tilde{v}} \end{array}\right]=\left(D-A_{1}\right)X-B_{1}\left[\begin{array}{c} \tilde{y}\\ -\eta^{\tilde{v}} \end{array}\right]+W_{X}^{\Theta }.$$

We now aim to mathematically prove that the (generalized) state and input observer of DEM can be reduced into an augmented LTI system, for which an exact discretization can be performed. We proceed by simplifying the mean field terms in Equation (Equation 39) as:(86)$$\begin{array}{cc} W_{X}^{\Theta }= & -\frac{1}{2}\left[\begin{array}{c} tr\left(\Sigma^{\theta }\left(N_{X^{1}}^{T}{\tilde{\Pi }}^{z}N+M_{X^{1}}^{T}{\tilde{\Pi }}^{w}M\right)\right)\\ tr\left(\Sigma^{\theta }\left(N_{X^{2}}^{T}{\tilde{\Pi }}^{z}N+M_{X^{2}}^{T}{\tilde{\Pi }}^{w}M\right)\right)\\ \ldots \end{array}\right]\left|{}_{\vartheta =\mu^{\vartheta }}\right.\\ = & -\frac{1}{2}\left[\begin{array}{c} vec{\left(\Sigma^{\theta T}\right)}^{T}\left(I⊗\left(N_{X^{1}}^{T}{\tilde{\Pi }}^{z}\right)vec\left(N\right)\right)+\\ \;vec{\left(\Sigma^{\theta T}\right)}^{T}\left(I⊗\left(M_{X^{1}}^{T}{\tilde{\Pi }}^{w}\right)vec\left(M\right)\right)\\ \ldots \end{array}\right]\left|{}_{\vartheta =\mu^{\vartheta }}\right.\\ = & \left(Z_{X}^{N}vec\left(N\right)+Z_{X}^{M}vec\left(M\right)\right)\left|{}_{\vartheta =\mu^{\vartheta }}\right., \end{array}$$
where,
(87)$$Z_{X}^{N}=-\frac{1}{2}\left(I⊗vec{\left(\Sigma^{\theta T}\right)}^{T}\right)\left[\begin{array}{c} I⊗(N_{X^{1}}^{T}{\tilde{\Pi }}^{z})\\ I⊗(N_{X^{2}}^{T}{\tilde{\Pi }}^{z})\\ \ldots \end{array}\right],\;Z_{X}^{M}=-\frac{1}{2}\left(I⊗vec{\left(\Sigma^{\theta T}\right)}^{T}\right)\left[\begin{array}{c} I⊗(M_{X^{1}}^{T}{\tilde{\Pi }}^{w})\\ I⊗(M_{X^{2}}^{T}{\tilde{\Pi }}^{w})\\ \ldots \end{array}\right].$$Since *M* and *N* can be obtained from linear transformation of *X*, vec(M) and vec(N) can be written as:(88)$$vec\left(M\right)=Z^{M}X\;\mathrm{and}\;vec\left(N\right)=Z^{N}X,$$
where $Z^{M}$ and $Z^{N}$ are matrices with elements 0 and 1. This leads to the mean field term being expressed as a linear transformation of *X*:(89)$$W_{X}^{\Theta }=\left(Z_{X}^{N}Z^{N}+Z_{X}^{M}Z^{M}\right)\left|{}_{\vartheta =\mu^{\vartheta }}\right.X.$$Substituting Equation (Equation 89) into Equation (Equation 85) simplifies the observer as:(90)$$\begin{array}{cc} \dot{X} & =A_{4}X+B_{4}\left[\begin{array}{c} \tilde{y}\\ -\eta^{\tilde{v}} \end{array}\right],\\ A_{4} & =D-A_{1}+\left(Z_{X}^{N}Z^{N}+Z_{X}^{M}Z^{M}\right)\left|{}_{\vartheta =\mu^{\vartheta }}\right.\;\mathrm{and}\;B_{4}=-B_{1}. \end{array}$$The (generalized) state and input observer given by Equation (Equation 90) is of the form of an augmented LTI system. Therefore, an exact discretization can be used to solve it without using the second order gradient $J^{X}$ as given in Equation (Equation 48). This reduces the algorithm’s computational complexity because $E_{XX}$ and $W_{XX}^{\Theta }$ for $J^{X}$ calculation are no longer necessary. Figure 3 shows the simplified control diagram of the observer. The stability condition of this observer (under known θ and λ) and its similarity with the Kalman Filter is discussed in our prior work [2]. To evaluate $W_{X}^{\Theta }$, one could either use Equation (Equation 89) or Equation (Equation 76). Equation (Equation 90) is derived mainly for simplification and exact discretization.

### 13.2. Parameter Estimation—System Identification

This section aims to mathematically prove that the E step can be reduced to an augmented LTI system, for which an exact discretization can be performed. We proceed by first simplifying the parameter update equation given in Equation (Equation 79):(91)$$\frac{\partial \Theta }{\partial a}=-P^{\theta }\left(\Theta -\eta^{\theta }\right)\left|{}_{\vartheta =\mu^{\vartheta }}\right.+\sum_{t}\left(-E_{\Theta }+W_{\Theta }^{X}\right).$$Grouping all $W_{\Theta^{i}}^{X}$ using Equation (Equation 72) yields:(92)$$\begin{array}{cc} W_{\Theta }^{X} & =-\frac{1}{2}\left(I⊗vec{\left(\Sigma^{XT}\right)}^{T}\right)\left[\begin{array}{c} I⊗{\tilde{\epsilon }}_{X\theta^{1}}^{T}\tilde{\Pi }\\ I⊗{\tilde{\epsilon }}_{X\theta^{2}}^{T}\tilde{\Pi }\\ \ldots \end{array}\right]vec\left({\tilde{\epsilon }}_{X}\right)\left|{}_{\vartheta =\mu^{\vartheta }}\right.=Z_{\theta }^{\epsilon }vec\left({\tilde{\epsilon }}_{X}\right)\left|{}_{\vartheta =\mu^{\vartheta }}\right. \end{array}$$
where
(93)$$\begin{array}{cc} {\tilde{\epsilon }}_{X\theta^{i}}^{T}\tilde{\Pi } & =-\left[\begin{array}{ccc} {\tilde{C}}_{\theta^{i}}^{T}{\tilde{\Pi }}^{z} & O & {\tilde{A}}_{\theta^{i}}^{T}{\tilde{\Pi }}^{w}\\ O & O & {\tilde{B}}_{\theta^{i}}^{T}{\tilde{\Pi }}^{w} \end{array}\right],vec\left({\tilde{\epsilon }}_{X}\right)=-vec\left[\begin{array}{cc} \tilde{C} & O\\ O & O\\ \tilde{A} & \tilde{B} \end{array}\right]+vec\left[\begin{array}{cc} O & O\\ O & I\\ D^{x} & O \end{array}\right]=-Z^{\theta }\theta +Z^{I}. \end{array}$$Here $Z^{\theta }$ and $Z^{I}$ are constant matrices with elements 0 and 1. Substituting $vec({\tilde{\epsilon }}_{X})$ from Equation (Equation 93) in Equation (Equation 92) yields:(94)$$W_{\Theta }^{X}=-\left(Z_{\theta }^{\epsilon }Z^{\theta }\right){{|}_{\vartheta =\mu^{\vartheta }}\Theta +\left(Z_{\theta }^{\epsilon }Z^{I}\right)|}_{\vartheta =\mu^{\vartheta }}.$$Substituting Equations (Equation 62) and (Equation 94) in Equation (Equation 91), simplifies the parameter update equation to
(95)$$\begin{array}{cc} \frac{\partial \Theta }{\partial a} & =A_{5}\Theta +B_{5},\\ A_{5} & =-\left[P^{\theta }+n^{t}\left(Z_{\theta }^{\epsilon }Z^{\theta }\right){|}_{\vartheta =\mu^{\vartheta }}+\sum_{t}A_{2}\right],\;B_{5}=P^{\theta }\eta^{\theta }+n^{t}\left(Z_{\theta }^{\epsilon }Z^{I}\right){|}_{\vartheta =\mu^{\vartheta }}+\sum_{t}B_{2}\left[\begin{array}{c} \tilde{y}\\ \mu^{\tilde{x}} \end{array}\right], \end{array}$$
where $A_{2}$ and $B_{2}$ are given in Equation (Equation 62). Equation (Equation 95) is a linear differential equation in Θ for which an exact discretization can be computed. For each Θ update in Algorithm 2, $A_{2}$ and $B_{2}$ are also updated, consequently updating $A_{5}$ and $B_{5}$. Therefore, Equation (Equation 95) is equivalent to a linear time-*varying* system. Figure 4 shows the simplified parameter estimation step of the robot brain. To evaluate $W_{\Theta }^{X}$, one could use either Equation (Equation 72) or Equation (Equation 94). Equation (Equation 94) was derived mainly for the exact discretization and for the convergence proof in Section 14.

### 13.3. Hyperparameter Update

The update equation in Equation (Equation 80) can be simplified as:(96)$$\begin{array}{cc} \frac{\partial \Lambda }{\partial b}= & k^{\Lambda }\left[-P^{\lambda }\epsilon^{\lambda }+\frac{n^{t}}{2}\left[\begin{array}{c} n^{{\tilde{\Pi }}^{z}}\\ n^{{\tilde{\Pi }}^{w}} \end{array}\right]-\frac{1}{2}\left[\begin{array}{c} e^{\lambda^{z}}tr\left(\left(S⊗\Omega^{z}\right)A_{3}\right)\\ e^{\lambda^{w}}tr\left(\left(S⊗\Omega^{w}\right)B_{3}\right) \end{array}\right]\right]\left|{}_{\vartheta =\mu^{\vartheta }}\right.\\ = & -\frac{k^{\Lambda }}{2}\left[\begin{array}{cc} tr(\left(S⊗\Omega^{z}\right)A_{3}) & O\\ O & tr(\left(S⊗\Omega^{w}\right)B_{3}) \end{array}\right]e^{\Lambda }-\left(k^{\Lambda }P^{\lambda }\right)\Lambda +\left(k^{\Lambda }P^{\lambda }\eta^{\lambda }+\frac{k^{\Lambda }n^{t}}{2}\left[\begin{array}{c} n^{{\tilde{\Pi }}^{z}}\\ n^{{\tilde{\Pi }}^{w}} \end{array}\right]\right)\\ = & a_{1}e^{\Lambda }+a_{2}\Lambda +a_{3}, \end{array}$$
where $a_{1},a_{2}$, and $a_{3}$ are constants that are independent of Λ. Since Equation (Equation 96) is *nonlinear* in Λ, an approximate discretization like the conventional Gauss–Newton update scheme given in Equation (Equation 48) should be used for the M step. In summary, the D and E steps follow an exact discretization, whereas the M step follows an approximate discretization.

## 14. Convergence Proof for Parameter and Hyperparameter Estimation

In robotics, it is important that learning algorithms provide a stable solution, especially when robot safety during operation is a concern. Therefore, a proof of convergence for DEM is important for its widespread use in robotics as a learning algorithm. However, the DEM literature lacks any such mathematical proof of convergence for the estimator. Therefore, this section aims at providing one for the parameter and hyperparameter estimation step on LTI systems.

Since the update equation given by Equation (Equation 95) is a linear differential equation, proving that $A_{5}≺O$ is sufficient to prove that Θ converges to a stable solution. Substituting the expression for $A_{2}$ from Equation (Equation 62) to the $A_{5}$ in Equation (Equation 95), yields:(97)$$A_{5}=-\left[P^{\theta }+n^{t}Z_{\theta }^{\epsilon }Z^{\theta }+\sum_{t}\left(N^{T}{\tilde{\Pi }}^{z}N+M^{T}{\tilde{\Pi }}^{w}M\right)\right]\left|{}_{\vartheta =\mu^{\vartheta }}\right..$$Since the prior precision matrix can be chosen to be positive definite, $P^{\theta }≻O$. It is straightforward to note from the expression for $A_{2}$ in Equation (Equation 62) that $\sum_{t}A_{2}≻O$, because ${\tilde{\Pi }}^{z}≻O$, ${\tilde{\Pi }}^{w}≻O\Rightarrow N^{T}{\tilde{\Pi }}^{z}N≻O$, and $M^{T}{\tilde{\Pi }}^{w}M≻O$. Therefore, the proof of convergence is complete if we prove that $Z_{\theta }^{\epsilon }Z^{\theta }≻O$. Simplifying the expressions for $Z_{\theta }^{\epsilon }$ and $Z^{\theta }$ from Equations (Equation 92) and (Equation 93), after some nontrivial linear algebra [41], yields:(98)$$Z_{\theta }^{\epsilon }Z^{\theta }=\frac{1}{2}{\frac{\partial \tilde{\theta }}{\partial \theta }}^{T}Z_{1}Z_{2}\frac{\partial \tilde{\theta }}{\partial \theta }$$
where
$$\begin{array}{cc} Z_{1}= & \;diag\left({\tilde{\Pi }}^{w}⊗I,{\tilde{\Pi }}^{w}⊗I,{\tilde{\Pi }}^{z}⊗I\right),\;\frac{\partial \tilde{\theta }}{\partial \theta }=\;diag\left(vec{\tilde{A}}_{vecA^{T}}^{T},vec{\tilde{B}}_{vecB^{T}}^{T},vec{\tilde{C}}_{vecC^{T}}^{T}\right),\\ Z_{2}= & \left[\begin{array}{ccc} I⊗\Sigma^{\tilde{x}\tilde{x}T} & I⊗\Sigma^{\tilde{v}\tilde{x}T} & O\\ I⊗\Sigma^{\tilde{x}\tilde{v}T} & I⊗\Sigma^{\tilde{v}\tilde{v}T} & O\\ O & O & I⊗\Sigma^{\tilde{x}\tilde{x}T} \end{array}\right]. \end{array}$$It is straightforward from Equation (Equation 98) that $Z_{\theta }^{\epsilon }Z^{\theta }≻O$ because $Z_{1}≻O$, and $Z_{2}≻O$. Combining all the results from this section, $P^{\theta }≻O$, $Z_{\theta }^{\epsilon }Z^{\theta }≻O$ and $\sum_{t}A_{2}≻O,\Rightarrow A_{5}≺O$. This completes the proof that the parameter estimation step of DEM converges for an LTI system. Similarly, from Equation (Equation 81), $J^{\Lambda }≺O$ proves the convergence of hyperparameter estimation step. For a detailed account of the linear algebra behind the proof of convergence, readers may refer to [41].

## 15. A Demonstrative Example

This section aims to provide the proof of concept for DEM through simulation for the estimation of an LTI system with colored noise. Since the algorithm can find an infinite number of solutions for a black box estimation of $\tilde{x}$, $\tilde{v}$, θ and λ from y, a black box estimation is not ideal as a demonstrative example. Therefore, we restrict this section to the joint estimation of *x*, *A*, *B*, $\Pi^{w}$, and $\Pi^{z}$ from known y and *C*.

### 15.1. Generative Model

A stable LTI system of the form Equation (Equation 3) was selected, with randomly generated parameters $\theta^{i}\in \left[-1,1\right]$ having
$$A=\left[\begin{array}{cc} 0.0484 & 0.7535\\ -0.7617 & -0.2187 \end{array}\right],B=\left[\begin{array}{c} 0.3604\\ 0.0776 \end{array}\right],C=\left[\begin{array}{cc} 0.2265 & -0.4786\\ 0.4066 & -0.2641\\ 0.3871 & 0.3817\\ -0.1630 & -0.9290 \end{array}\right].$$A Gaussian bump input signal of $v=e^{-0.25{\left(t-12\right)}^{2}}$ was centered around t=12s and sampled at dt=0.1s till T=32s was used. The colored noise was generated with a smoothness value of σ=0.5 for the Gaussian kernel. The noise precisions were $\Pi^{w}=e^{8}I_{2}$ and $\Pi^{z}=e^{8}I_{4}$, making $\lambda^{z}=\lambda^{w}=8$. The embedding order of the generalized motion of states and inputs were p=6 and d=2, respectively.

### 15.2. Priors for Estimation

As discussed in Section 5.2, three prior distributions are necessary for the algorithm. Since the inputs are known, the input prior $\eta^{v}$ is initialized with the known input *v*, and a tight prior precision of $P^{v}=e^{32}I_{1}$ is used to restrict any changes in *v*. Similarly, since the parameter *C* is known, the corresponding prior parameters in $\eta^{\theta }$ are initialized with *C*, with tight priors of $P^{\theta^{i}}=e^{32}$. The prior parameters $\eta^{\theta^{i}}$ for the unknown *A* and *B* matrices are randomly sampled from the range of [−2,2], and a low prior precision of $P^{\theta^{i}}=e^{6}$ is used to encourage exploratory behavior. In summary, $\eta^{\theta }={\left[\begin{array}{ccc} vec{\left(rand{\left(2,2\right)}^{T}\right)}^{T} & vec{\left(rand{\left(2,1\right)}^{T}\right)}^{T} & vec{\left(C^{T}\right)}^{T} \end{array}\right]}^{T}$ and $P^{\theta }=diag\left(e^{6}I_{4},e^{6}I_{2},e^{32}I_{8}\right)$. Since the hyperparameters are unknown, their priors were set to zero $\lambda ={\left[0\;0\right]}^{T}$, with a prior precision of $P^{\lambda }=e^{3}I_{2}$ to encourage exploration.

### 15.3. Results of Estimation

The data y generated from the system in Section 15.1 was used to run the DEM algorithm given in Algorithm 2. Figure 5a demonstrates the successful state estimation of the algorithm. The results of parameter estimation (*A* and *B*) are shown in Figure 5b. The updates began from randomly selected priors $\eta^{\theta }$, marked by red circles, to finally converge. Table 1 shows that the DEM’s estimate of *A* and *B* are close to the real values.

This confirms that the parameter estimation can converge close to the real parameters, even when $\eta^{\theta }$ is randomly selected from the range $\eta^{\theta }\in \left[-2,2\right]$ that is double the size of the real parameter range $\theta^{i}\in \left[-1,1\right]$. Figure 5c shows the successful hyperparameter convergence close to $\lambda^{z}=\lambda^{w}=8$.

DEM’s confidence on its estimates increase with the E step iterations, as can be seen from Figure 6a, which demonstrates an increase in parameter precision $\Pi^{\theta }$. A similar trend can be observed for $\Pi^{X}$. However, $\Pi^{\lambda }$ remains a constant during the entire algorithm, as proved in Section 12.2. The key idea behind DEM’s inference is the maximization of free energy objectives. Read together, Figure 5 and Figure 6 demonstrates that DEM successfully estimates $\tilde{x}$, θ and λ, with increasing confidence on its estimates as the estimation proceeds by maximizing $\bar{F}$ from Equation (Equation 40). In summary, DEM can be used for the joint estimation of states, parameters and hyperparameters of an LTI system, subjected to colored noise.

## 16. Benchmarking

This section deals with benchmarking DEM against the state-of-the-art parameter estimation methods such as Expectation Maximization (EM), Subspace method (SS), and Prediction Error Minimization (PEM), for black-box estimation (fully unknown *x*, θ and λ).

### 16.1. Evaluation Metric for Parameter Estimation

For the black box identification, with completely unknown *x*, θ, and λ, there are infinite solutions with accurate input–output mapping. However, for LTI systems, there exists a unique transformation for identical systems. We use the companion canonical form to check the validity of parameter estimation by transforming both the real and the estimated parameters into their companion canonical form and then using the (square of) Euclidean distance between them as the sum of squared error (SSE) in parameter estimation. This evaluation metric will be used for parameter estimation in the next section.

### 16.2. Simulation Setup

A total of 500 (5 × 100) different randomly generated stable systems were used with five different noise smoothness values for parameter estimation. All systems were selected with same number of parameters $n^{\theta }=14$ (n=2,m=4 and r=1), with each $\theta^{i}\in \left[-1,1\right]$, while ensuring that *A* matrix is stable. All the noises were generated with the precision of $e^{6}\left(\Pi^{w}=e^{6}I_{2\times 2},\Pi^{z}=e^{6}I_{4\times 4}\right)$, with the embedding orders of states and inputs as p=6 and d=2. A Gaussian bump of $v=e^{-0.25*{\left(t-12\right)}^{2}}$ was used as the input signal with dt=0.5s and T=32s. The prior parameter was randomly initialized such that all $\eta^{\theta^{i}}\in \left[-2,2\right]$ with a tight prior precision of $P^{\theta }=e^{4}I_{14\times 14}$. Both the hyperparameter priors $\eta^{\lambda^{i}}$ were set to zero, with a prior precision of $P^{\lambda }=e^{-4}I_{2\times 2}$.

The System Identification toolbox from MATLAB was used for SS (n4sid()) and PEM methods. The solution of SS was used to initialize PEM. An implementation of EM algorithm for state space models was written in MATLAB based on [45]. n4sid() is inherently designed to handle colored noise, whereas the implemented EM algorithm is not. The code for the DEM algorithm will be openly available at: https://github.com/ajitham123/DEM_LTI.

### 16.3. Results

The results shown in Figure 7 demonstrate the superior performance of DEM in comparison with EM, PEM, and SS, with minimum SSE during parameter estimation across different noise smoothness. Additionally, EM and PEM exploded occasionally (<5% times), resulting in outliers in SSE, which were removed for better visualization. DEM demonstrated a consistent performance without generating any such outliers or exploding solutions, which could be explained by DEM’s convergence guarantees for parameter estimation under colored noise [41], as proved in Section 14. In summary, DEM is a competitive parameter estimator for LTI systems with colored noise.

## 17. Discussion

The quest for a brain-inspired learning algorithm for robots has culminated in the free energy principle that postulates biological brain’s perception as an optimization over its free energy objectives. FEP is of prime importance to robotics because of the use of generalized coordinates that enables it to gracefully handle colored noises. Colored noises appear in real robotics systems through the unmodeled dynamics and the non-linearity errors in the model, thereby providing an advantage for DEM during estimation when compared to other estimators. An example could be the unmodeled wind disturbances acting on an unmanned aerial vehicle while in flight, or the non linearity errors in the dynamic model of a robotic manipulator arm involved in a pick and place operation. The scope of this work spans across the blind system identification of such linear dynamic systems with colored noise.

The fundamental difference between this work and the prior work is in the reformulation of DEM for an LTI system. While DEM from computational neuroscience focuses on emulating the biological brain’s perception through the hierarchical abstraction of a number of non-linear dynamic systems that interact with each other, our work focuses on reducing this method into an algorithm for the system identification of an LTI system with colored noise, which is a well-known problem in robotics. This reformulation enables the standard analysis for convergence, stability and unbiased estimation, which is an essential analysis in practical robotics. It also enables DEM to be compared with other existing estimation algorithms in a control systems domain. The widespread use of DEM in robotics necessitates these mathematical analyses, especially when concerning the stable and safe operation of robots in industry and during human–robot interaction.

An algorithm with proved convergence for estimation is preferred for safe robotic applications. Therefore, one of the main contribution of this work was the reduction of the estimation algorithm into a coupled augmented system to prove the convergence of parameter and hyperparameter estimation steps. This work also demonstrated the successful applicability of DEM for the estimation of a randomly selected LTI system. Furthermore, we showed through rigorous simulations on a wide range of randomly generated LTI systems that DEM is a competitive algorithm for system identification under colored noise, thereby widening the scope of DEM to a large number of LTI systems in robotics.

One of the main drawbacks of the algorithm is its higher computational complexity when compared to the estimation algorithms that do not keep track of the trajectory of states. Therefore, future work can focus on the online estimation using DEM with reduced computational load. Future work can also focus on extending this algorithm for linear time varying systems to deal with robots with changing system parameters while in operation—a delivery drone dropping deliveries in mid-flight, for example. From a practical robotics point of view, DEM’s parameter estimation module can be directly applied to a wide range of robots such as quadrotors, robotic arms, wheeled robots, etc. for black-box system identification, the input estimation module can be employed for fault-detection systems, and the hyperparameter estimation module can be used for online noise estimation for robust control. DEM can also be extended with a control loop for active inference to perform simultaneous perception and action on robots. This would result in the development of cognitive robots that can learn the generative model in the environment by interacting with it and actively seeking new information (active learning) for uncertainty resolution. This would influence multiple domains in robotics such as human–robot interaction for task learning, swarm robotics for collective learning and distributed control, informative path planning of aerial robots for environment monitoring, etc. The development of such brain-inspired autonomous agents sits at the core of cognitive robotics research. In summary, DEM has a huge potential to be the bioinspired learning algorithm for future robots.

## 18. Conclusions

The free energy principle from neuroscience has a great potential to be one of the most prominent frameworks for learning and control for the autonomous systems in future. Therefore, this paper converted the FEP-based inference scheme called DEM into a joint state, input, parameter, and hyperparameter estimation algorithm for LTI systems with colored noise. We derived the mathematical framework of DEM for LTI systems to prove that the resulting estimator is a combination of linear estimators that are coupled. We provided the proof of convergence for the estimation steps. Through rigorous simulations on randomly generated linear systems with colored noise at varying smoothness levels, we demonstrated that the DEM algorithm outperforms EM, PEM, and SS methods for parameter estimation with minimal estimation error. In light of the potential for DEM to solve the parameter estimation problem, the future research will aim at applying DEM to a quadcopter flying in wind.

## Figures and Tables

**Figure 1 entropy-23-01306-f001:**
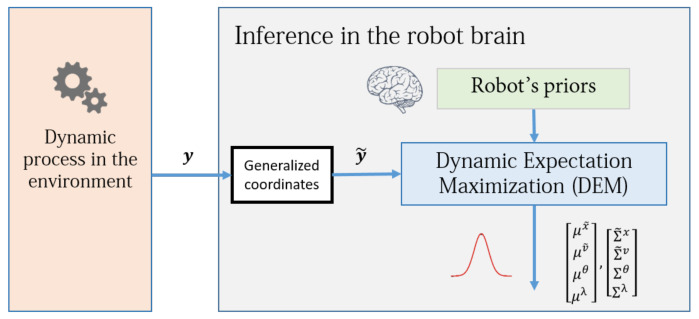
A simple block diagram of the robot brain’s inference process using DEM. It uses the measurement data y generated from the environment (also called generative process). DEM enables the direct fusion of the prior information into the inference process. The concept of generalized coordinates will be detailed in Section 3.1.

**Figure 2 entropy-23-01306-f002:**
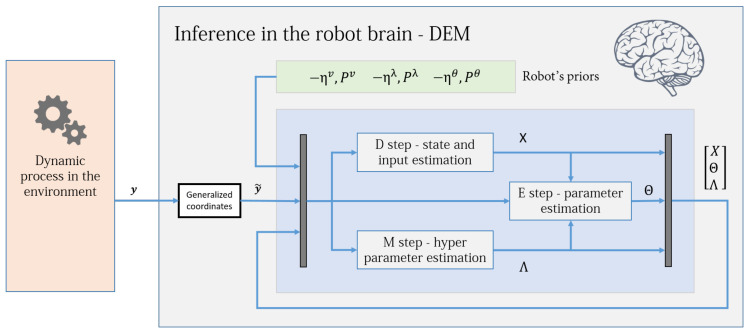
The DEM algorithm is represented using three coupled steps: D, E, and M steps. The algorithm combines the data from the environment with the robot’s prior beliefs to infer the states, inputs, parameters and hyperparameters of the system. For each parameter update in the E step, the D step updates the (generalized) states and inputs for all times instances, and the M step iterates until hyperparameter convergence, as demonstrated in Algorithm 1. The dynamic process is the generative model in Section 3.1, the priors are the distributions given in Section 5.2 and the generalized coordinates block is defined in Section 3.4. Section 13 will elaborate on the D, E, and M blocks.

**Figure 3 entropy-23-01306-f003:**
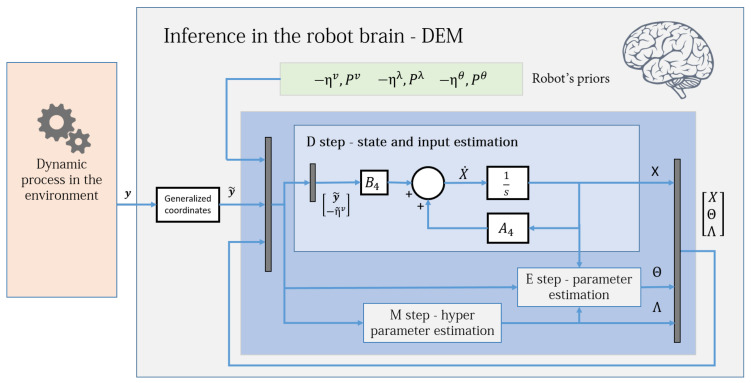
The DEM algorithm for an LTI system, with the D step simplified as an augmented LTI system given by Equation (Equation 90). The D-step block corresponds to the D-step loop in Algorithm 2 and operates at a different frequency from the E and M blocks.

**Figure 4 entropy-23-01306-f004:**
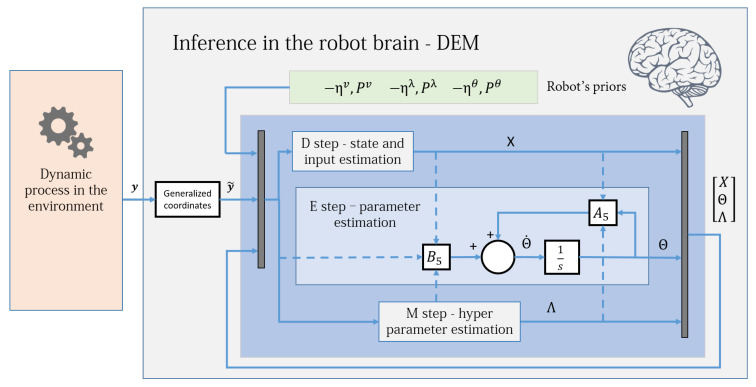
The DEM algorithm for an LTI system, with the E step simplified as an augmented LTI system given by Equation (Equation 95). The E-step block corresponds to the E-step outer loop in Algorithm 2 and operates at a different frequency when compared to the D and M blocks. The dotted lines illustrate the flow of variables from other blocks and demonstrate the coupled nature of D, E, and M steps. This diagram is illustrative and should not be confused with a control diagram.

**Figure 5 entropy-23-01306-f005:**
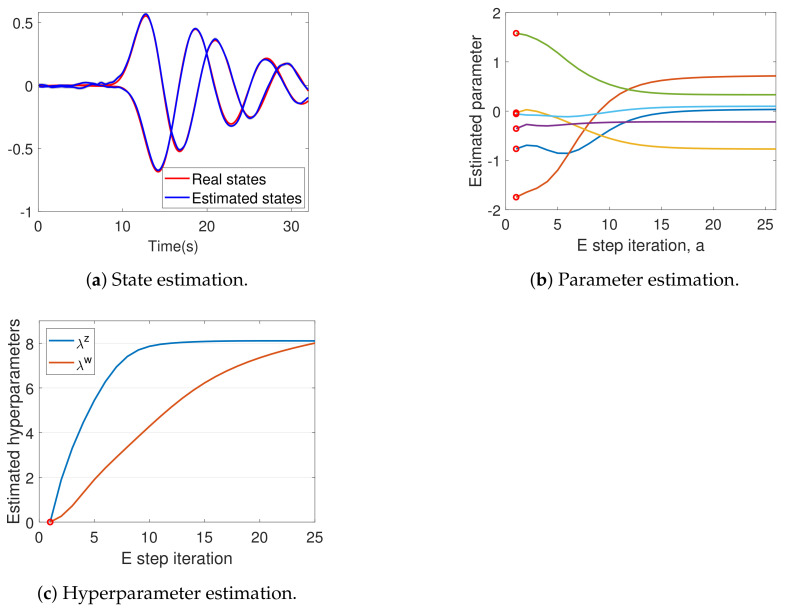
The results of DEM’s estimation process. (**a**) The estimated states in blue closely resembles the real states in red. (**b**) The parameter estimation starts from randomly selected $\eta^{\theta }$, marked by red circles and converges with each E step iteration *a*. (**c**) Both the hyperparameters start from $\eta^{\lambda }=0$, and converge close to the correct value of 8.

**Figure 6 entropy-23-01306-f006:**
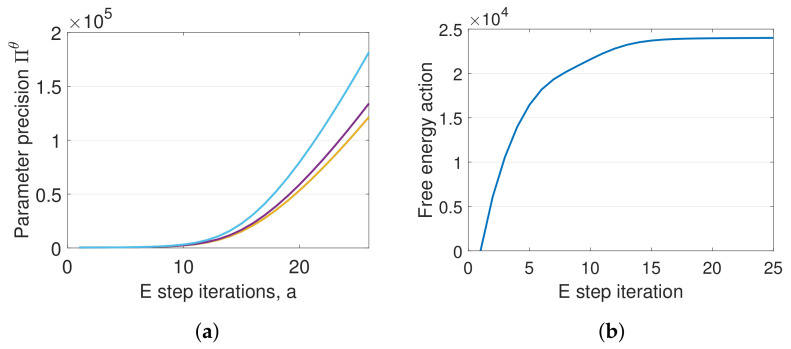
Maximization of $\bar{F}$ improves the confidence on estimates. (**a**) Parameter precision $\Pi^{\theta }$. (**b**) Free action $\bar{F}(a)-\bar{F}(0)$.

**Figure 7 entropy-23-01306-f007:**
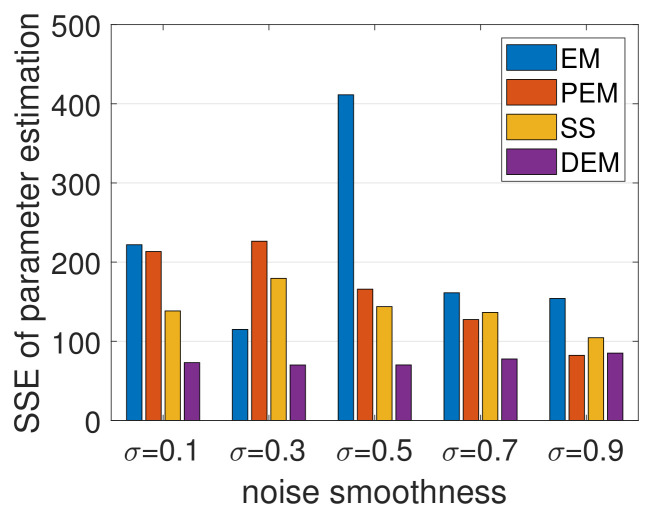
The sum of all SSE of Θ for 100 random systems each, for 5 different noise smoothnesses. DEM outperforms EM, PEM, and SS with minimum SSE for parameter estimation under colored noise.

**Table 1 entropy-23-01306-t001:** DEM’s estimate of *A* and *B* converges to real value.

	$\theta_{1}$	$\theta_{2}$	$\theta_{3}$	$\theta_{4}$	$\theta_{5}$	$\theta_{6}$
Real	0.048	0.753	−0.761	−0.218	0.360	0.077
Estimate	0.034	0.714	−0.769	−0.219	0.333	0.098

## Data Availability

The MATLAB code for the DEM algorithm will be openly available at: https://github.com/ajitham123/DEM_LTI.

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
