# Peer review of "Dynamic Expectation Maximization Algorithm for Estimation of Linear Systems with Colored Noise"

_entropy, 2021, doi:10.3390/e23101306_

Round 1

Reviewer 1 Report

Meera and Wisse presents a treatment of linear systems with colored noise. They evaluate the usefulness of dynamic expectation maximization algorithm, an extension of variational expectation maximization algorithm which was proposed by Karl Friston to handle inversion of dynamic causal models with non-linear state space in continuous time. The paper is very well written and provides novel reformulations to deal with linear time invariant systems with colored noise. This is done in the context of robotics which is a fascinating application of active inference algorithms. I don’t have any major concerns. Nevertheless, I list below a few points that may help to improve the paper:

  • DEM is presented as a variant of FEP. I think this is not the right way to position these. FEP is a framework, a normative principle, while DEM is an algorithm which FEP may employ for inference in dynamic settings. This needs to be clarified throughout the manuscript. If DEM is a variant, then is a variant of active inference not FEP.
  • FEP is described as a unified theory of brain, which is historically/ chronologically correct, but one can be more precise: FEP is the unified account of how brain implements various functions. However, more recent expositions have even moved beyond brains towards any system that survives over time including societies and cultures.
  • Almost all neuroimaging literature, where DEM has it roots, considers v to be known then why they are considered to be part of the parameters space \nu = {x’, v, \theta, \lambda}? For e.g., in brain imaging analyses, v represents known audio or visual stimulus (or causes). Initially I tried to take it in the context of robotics, but most of the preliminary material is present from the brain’s perspective, so it is little puzzling. One can simplify the expressions by dropping v without any profound implications. This is unless that input v also has uncertainties attached to them. However, I am unsure if only Gaussian causes makes much sense, this is going to be very limiting, and I am unsure how non-Gaussian causes can be tackled (which will violate Laplace approximation)?
  • It is not clear, how one can account for the variation (i.e., uncertainty) in hyperparameters which may be an important consideration in the present context.
  • Can you briefly comment if one can extend the notion of generalized coordinates to not only cover the space but also time?
  • Here the mean-field approximation is considered, i.e., time variant states and time invariant parameters and hyperparameters are considered conditionally independent. While this is a very useful assumption to simplify lots of math, dropping this assumption is also possible as has been demonstrated in the formulation provided in the ‘generalized filtering’ framework [1]. This can be important to recover the causal structure of data (see example given in [1, Section 5]). Some comments on this approach will be useful and how this can be incorporated in the current formalism with colored noise.

[1] Friston K, Stephan K, Li B, and Daunizeau J. Generalised Filtering, Mathematical Problems in Engineering. vol. 2010, Article ID 621670, 34 pages, 2010.

Reviewer 2 Report

The authors have presented a novel (and simplified) formulation of Dynamic Expectation Maximization (DEM) for linear time-invariant (LTI) systems. This formulation extends the applicability of DEM for use in LTI systems and robotics by overcoming the white-noise assumption that is commonly used in practise.

The authors have shown proof of convergence of state and parameter estimation under reasonable assumptions and provide valid benchmarks against state-of-the art parameter estimation algorithms used in the control system literature. The formulation provided in Section 13 will be useful for both control engineers and those doing research with DEM in neuroscience and machine learning.

I have found no errors in the mathematical derivations or formulations in the manuscript.

A minor point, but the use of generalised coordinates is predicated on the assumption that the noise processes within the model are analytic. Noise has a significant role in DEM and it’s claims of biological plausibility are centered primarily around the optimisation of the hyperparameters. The authors may wish to cite the following paper that addresses the significance of noise in solving the continuous-discrete filtering problem:

Balaji, Bhashyam, and Karl Friston. "Bayesian state estimation using generalized coordinates." Signal Processing, Sensor Fusion, and Target Recognition XX. Vol. 8050. International Society for Optics and Photonics, 2011.

The phrase ‘priors of the brain’ is used on several occasions; to someone unfamiliar with the free energy / Bayesian brain literature this may be confusing. I was also surprised to see that the first mention of ‘prediction error’ on 258. Perhaps the term ‘priors in the brain’ could be clarified in the introduction within the context of predictive coding.

32: It is one particular implementation of, rather than a ‘variant’ of the FEP.

57: DEM is a variational inference method. Please clarify the distinction being made here.

96: Please clarify or this statement; I do not have a control-systems background but I was able to identify a few papers in robotic control that utilise generalised coordinates in their methodology.

110: block derivative operation?

172: Suggest using ‘free action’ rather than ‘free energy action’

195: This is the key idea behind variational inference, not just DEM.

227: DEM as a triple estimation scheme relies upon this factorisation. This significance of this assumption seems understated here. It would be useful to explain why this factorisation is necessary (i.e separation of timescales).

271: This claim may be misinterpreted; a large mismatch may result in a large prediction error signal, but due to precision weighting this may not necessarily induce a large change in free energy.

421: Jacobians

700: This is an interesting result. Perhaps explain exactly why DEM is consistent while the others are not.

744: This paper does a good job in presenting DEM to an audience in robotics/control systems, as such it may be worth discussing ADEM (the active inference extension of DEM) in more detail in the discussion, as this may be appealing to the readership.

Reviewer 3 Report

The authors in this paper revisit the method known as dynamic expectation maximisation and present a detailed application to linear random dynamical systems.

On one hand the paper does not bring sufficient novelty, in comparison to past publications on DEM, to warrant a standalone article. Linear dynamical systems have been included in the most of publications on DEM, at least as a benchmark example before presenting more interesting non-linear problems. Hence, I feel that the paper has to be presented differently, either as a technical note, which unpacks lots of details of DEM, or as a tutorial with coding examples with implementations of the part of DEM.

I think it is valuable the authors effort to provide clarity in this domain, specially for researchers in other fields, such as robotics, which seems to be their goal. However, if that is the case than I am a bit confused by often references in the article to the neuronal networks, brain priors and similar. If the goal of the authors is to unpack the method, then one should also use a terminology which is appropriate to that field. For example, authors start with introducing linear plant system (LTI), a term which is rarely used in neuroscience (if at all), and in the rest of the paper they present figures and terminology which is purely neuroscience motivated. This discrepancy is quiet confusing and does not match the motivation described in the introduction.

A final general comment would be that the authors can describe the method in quite more detail. They are still only skimming over important parts, which are relevant for the implementation (e.g. how is time integral over the free energy computed) and focus mostly on update equations, which in my opinion are trivial to implement with modern probabilistic programing languages, and do not require such detailed unpacking.

In what follows I list some of the concerns with the paper I have and potential errors  I found in some parts:

  1. Eq. 8 does not seem correct to me. If one would do a Taylor expansions of the terms on the left hand side of the equations one would get different coefficients than what is shown in the ϒ matrix on the right hand side. For example y(t+dt) = y(t) + y'(t)dt + y''(t) dt2/2 + ..., similarly y(t-dt) = y(t) - y'(t)dt + y''(t)dt2/2. Note that y(t) = y(t) hence one row of the ϒ matrix has to correspond to (1, 0, 0, ...).
  2. The Eq 20 does not present a probability distribution with respect to \tilde{x}. If we are interested in conditional distribution of p(\tilde{x}|...) than one would have to express Eq 3 first in terms of \tilde{x} = (D - A)-1(...) before being able to express the probability distribution of \tilde{x} variable.
  3. The authors should clarify throughout the paper a difference between a time point, e.g. x(t), and a trajectory (x(t), x(t+dt), ...). It is not clear when they are talking about one and when about the other.
  4. Strangely, the authors suddenly introduce a variable X = (\tilde{x}, \tilde{v}) (after eq 33) and in subsequent equations they compute derivatives of free energy with respect to X. This is quite wrong, \tilde{x} and \tilde{x} are latent variables, the variational free energy is not a function of those variables, but of the sufficient statistics of the approximate posterior, e.g. q(\tilde{X}| \lambda). Where the gradient descent is computed with respect to \lambda (mean and the covariance matrix of the posterior), rather than hidden states themselves.
  5. Past publications of DEM did an extensive comparison of DEM to other Bayesian methods, such as Kalman Filtering and smoothing. What is the motivation here of comparing DEM to non-Bayesian approaches. Surely, in problems with lots of noise and uncertainty on the parameters, they will not be able to provide good results, even in linear systems. This is not a surprising finding given years of works on Bayesian statistics.

Round 2

Reviewer 1 Report

Authors have addressed my concerns.

Author Response

The reviewer did not have any further comments.

Reviewer 2 Report

The authors have addessed all of my concerns and made the appropriate changes/additions. 

Author Response

(The authors gave the same response as above.)
